# Evolutionary signatures of human cancers revealed via genomic analysis of over 35,000 patients

Diletta Fontana [1,8], Ilaria Crespiatico[1,8], Valentina Crippa[1,8], Federica Malighetti [1], Matteo Villa [1], Fabrizio Angaroni[2,3], Luca De Sano [2], Andrea Aroldi[1,4], Marco Antoniotti [2,5], Giulio Caravagna[6], Rocco Piazza [1], Alex Graudenzi [2,5,7] ✉, Luca Mologni [1] & Daniele Ramazzotti [1] ✉

Recurring sequences of genomic alterations occurring across patients can highlight repeated evolutionary processes with significant implications for predicting cancer progression. Leveraging the ever-increasing availability of cancer omics data, here we unveil cancer's evolutionary signatures tied to distinct disease outcomes, representing "favored trajectories" of acquisition of driver mutations detected in patients with similar prognosis. We present a framework named ASCETIC (Agony-baSed Cancer EvoluTion InferenCe) to extract such signatures from sequencing experiments generated by different technologies such as bulk and single-cell sequencing data. We apply ASCETIC to (i) single-cell data from 146 myeloid malignancy patients and bulk sequencing from 366 acute myeloid leukemia patients, (ii) multi-region sequencing from 100 early-stage lung cancer patients, (iii) exome/genome data from 10,000+ Pan-Cancer Atlas samples, and (iv) targeted sequencing from 25,000+ MSK-MET metastatic patients, revealing subtype-specific single-nucleotide variant signatures associated with distinct prognostic clusters. Validations on several datasets underscore the robustness and generalizability of the extracted signatures.

Cancer development is a stochastic evolutionary process that involves large populations of cells. Random genetic and epigenetic alterations that commonly occur in any cell can occasionally be beneficial to neoplastic cells, leading to the selection of clones with increased proliferation and survival abilities, eventually resulting in invasion and metastasis[1,2]. However, not all alterations are involved in this process, but only a relatively small subset of them, known as drivers, while most mutations are neutral, called passengers[3,4].

Building on the ever-increasing availability of omics data collected at various resolutions from NGS experiments on cancer patients, and on the continuous advances in cancer data science and machine learning, we are now empowered to explore the existence of cancer (sub)type-specific evolutionary signatures associated with different disease outcomes. These signatures represent the "favored trajectories" of driver alterations acquisition during cancer evolution that are repeatedly detected in patients with similar

[1]Department of Medicine and Surgery, University of Milano-Bicocca, Monza, Italy. [2]Department of Informatics, Systems and Communication, University of Milano-Bicocca, Milan, Italy. [3]Center of Computational Biology, Human Technopole, Milano, Italy. [4]Hematology and Clinical Research Unit, Fondazione IRCCS San Gerardo dei Tintori, Monza, Italy. [5]Bicocca Bioinformatics, Biostatistics and Bioimaging Centre—B4, Milan, Italy. [6]Department of Mathematics and Geosciences, University of Trieste, Trieste, Italy. [7]Institute of Molecular Bioimaging and Physiology, Consiglio Nazionale delle Ricerche (IBFM-CNR), Segrate, Milan, Italy. [8]These authors contributed equally: Diletta Fontana, Ilaria Crespiatico, Valentina Crippa. ✉e-mail: alex.graudenzi@unimib.it; daniele.ramazzotti@unimib.it

prognosis and can be exploited to stratify (unseen) patients accordingly.

The study of evolutionary signatures may allow us to delve into whether recurring genomic evolution patterns observed in cancer patients are consistently associated with improved or worsened prognoses. The primary goal of this approach is to improve predictive accuracy by extending the analysis beyond individual genetic alterations and by investigating the presence of these evolutionary trends. Through a comprehensive analysis of the broader spectrum of genomic alterations and their interactions, we can attain a more comprehensive understanding of cancer evolution and its influence on prognosis. This methodology strives to enhance predictive models by encompassing the intricate dynamics and interplay among genetic alterations, thus surpassing the limitations of considering individual genetic modifications.

In this work, we define the notion of cancer (sub)type-specific single nucleotide variants (SNV) evolutionary signatures associated to clusters of patients exhibiting statistically significant differences in prognosis. The concept of SNV evolutionary signatures is complementary to that of single base substitutions (SBS) mutational signatures, which were first introduced by Stratton and colleagues[5]. SBS mutational signatures aim to distinguish between the various

mutagenic mechanisms responsible for all the alterations that occur in cancer genomes, including both driver and passenger mutations. In contrast, the SNV evolutionary signatures introduced in this work identify conserved patterns of functionally advantageous alterations (i.e., driver mutations) that emerge from the complex interplay of multi-scale processes underlying cancer development and evolution. These signatures describe evolutionary steps that are shared among patients with similar prognosis. Here, we present a framework called ASCETIC (Agony-baSed Cancer EvoluTion InferenCe), that can process data from both bulk and single-cell sequencing experiments. Our method enables us to first reconstruct statistically robust models of tumor evolution for individual patients (Fig. 1). We then combine these individual models into a unique cancer-specific model of repeated evolution, in which we identify the significant evolutionary patterns, or evolutionary signatures, that are associated to outcome, by exploiting prognostic data via regularized Cox regression (Fig. 2). Moreover, our approach allows us to naturally combine genomic data from various resolutions and technologies, including bulk and single/multiple biopsies and single-cell sequencing experiments. We employ the ASCETIC framework to analyze SNV from several cancer datasets, including (1) single-cell sequencing data obtained through the Tapestri Platform from 146 patients with distinct myeloid malignancies and an

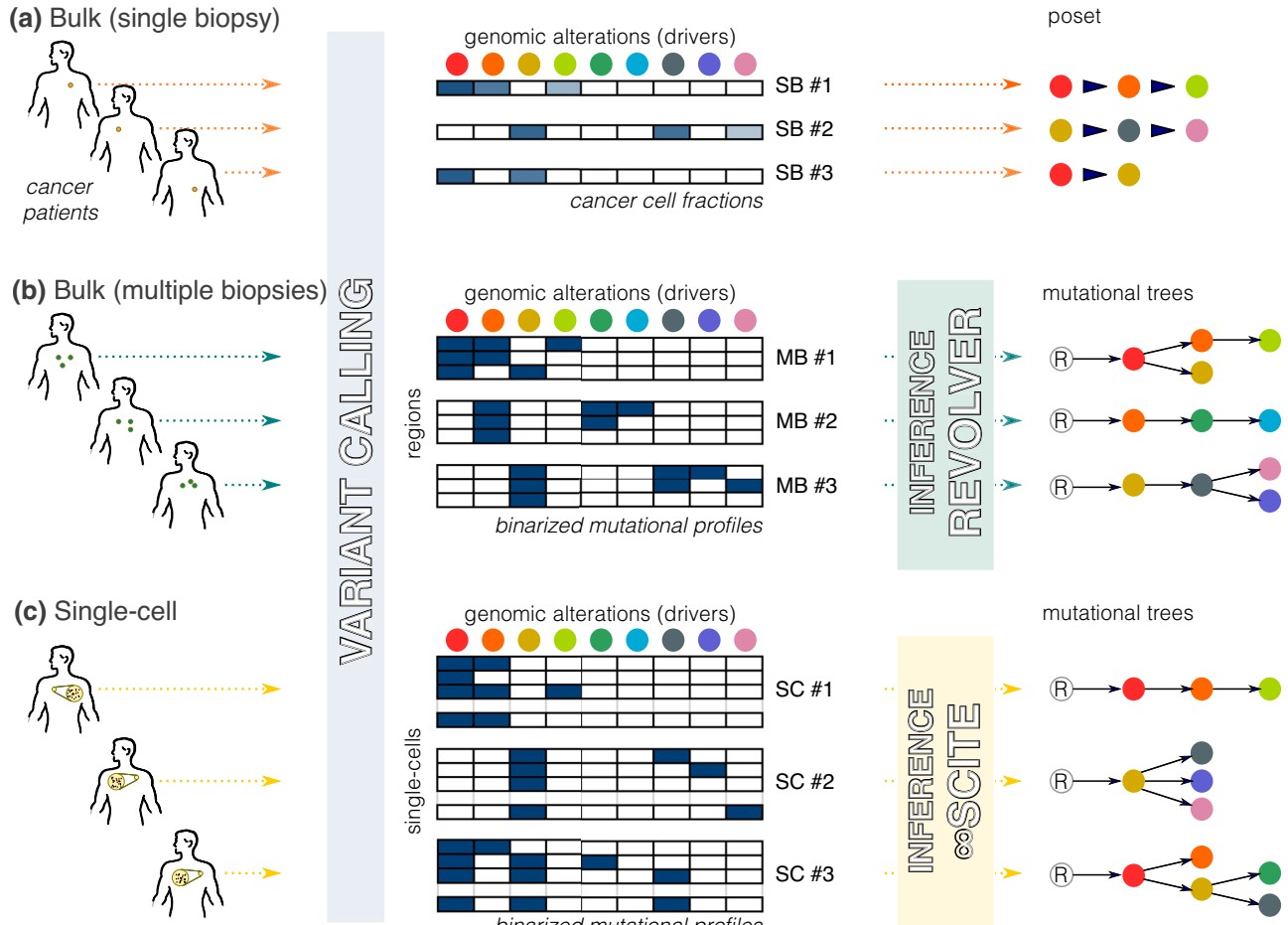

**Fig. 1 | ASCETIC pre-processing phase.** ASCETIC can efficiently process data at various resolutions, including the classical bulk NGS samples (**a**) providing a single biopsy per patient, multi-region sequencing data (**b**), as well as single-cell sequencing data (**c**). The framework includes an initial variant calling step, which can be performed adopting best practices for the considered sequencing technology. After this step, a set of mutational profiles are generated, comprising one sample per patient in the case of classical bulk NGS data, or multiple samples in the case of multiregion or single-cell data. From these inputs, ASCETIC first generates a set of temporal models of the evolution of each patient considered individually. For classical bulk NGS data, this is performed from cancer cell fractions, in order to obtain a partially ordered set (poset) for the set of driver mutations observed in each patient. For multi-region and single-cell data, ASCETIC exploits phylogenetic approaches tailored to analyze cancer data[6,65,66,71,72] in order to generate a mutational tree per patient. In this work, we used REVOLVER[6] for multi-region data and ∞SCITE[66] for single-cell data, although any preferred state-of-the-art algorithms can be used. The generated mutational profiles and evolutionary models are given as inputs for ASCETIC next steps.

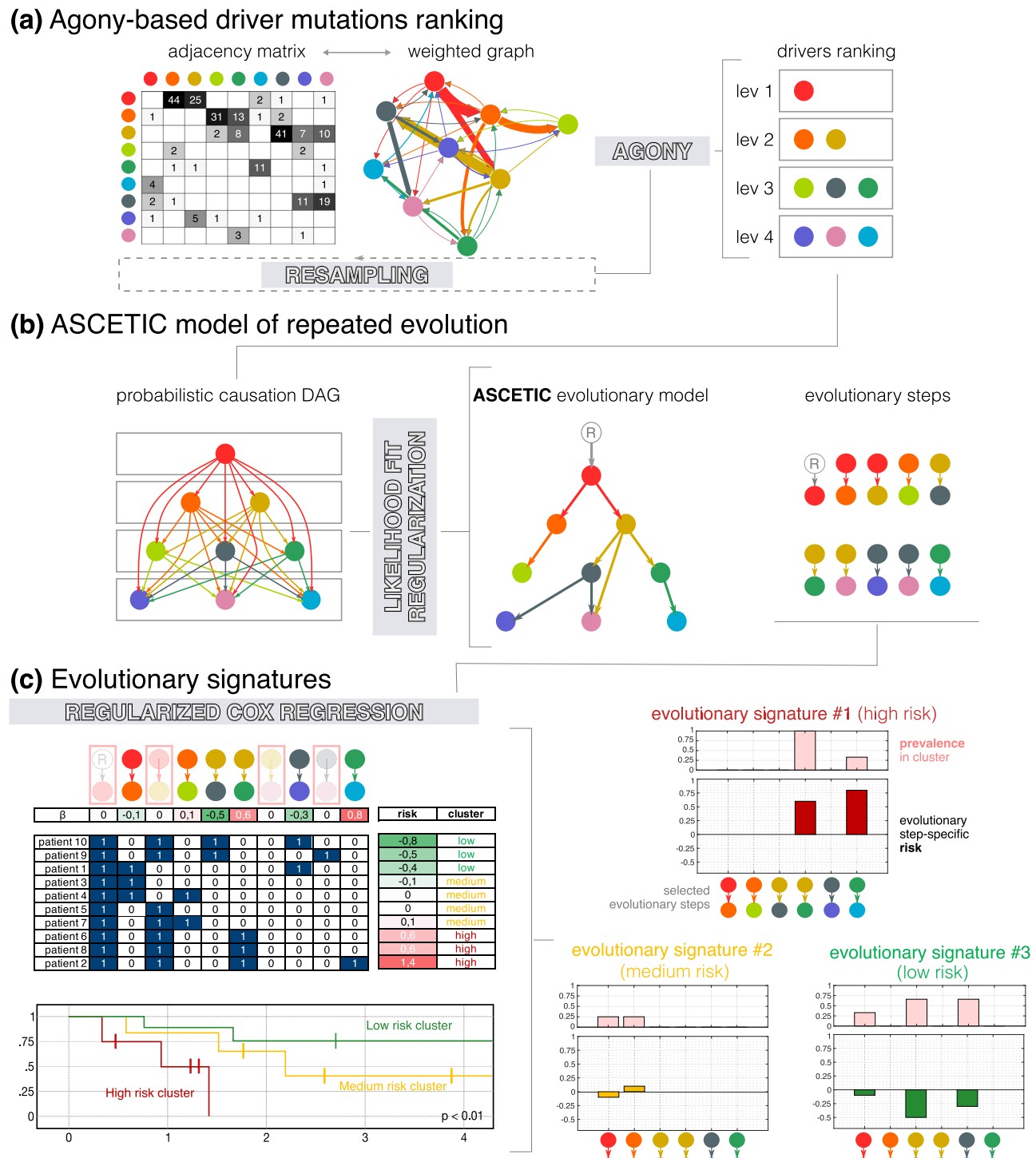

**Fig. 2 | ASCETIC decomposes the inference problem into three main tasks. a** First, it combines all the evolutionary models obtained during the pre-processing phase in order to build an agony-derived ranking[8] of the considered alterations, which implies a (partial) temporal ordering among these drives during cancer evolution. **b** Then, it adopts a likelihood-based approach grounded on the theory of probabilistic causation[9] for model selection in order to display the most significant relationships among driver mutations into a Bayesian Network depicting repeated evolutionary trajectories. **c** Finally, ASCETIC considers all the inferred evolutionary steps and exploits them to perform patients' stratification. This is done by considering survival data and selecting the most relevant features from the ASCETIC model of repeated evolution to stratify the samples into different risk groups or clusters. Survival analysis of the different risk groups is then performed via the Kaplan–Meier estimate. ASCETIC yields two main outputs: (1) a model that captures the consistent evolutionary trajectories observed across diverse patients throughout tumor evolution, and (2) a set of genomic features referred to as evolutionary signatures, which demonstrate significant associations with clinical outcomes. ASCETIC outputs inferred risk levels for each selected evolutionary step and display their relative prevalence within the cluster.

external dataset providing bulk whole exome sequencing data from 366 acute myeloid leukemia patients, (2) multi-region sequencing data from 100 early-stage lung cancer patients from the TRACERx project, (3) whole exome/genome sequencing data from over 10,000 Pan-Cancer Atlas samples, and (4) target bulk sequencing data from more than 25,000 MSK-MET metastatic patients (both datasets encompassing various cancer types). Furthermore, we conducted multiple validations of the evolutionary signatures extracted by ASCETIC using diverse and previously unseen datasets. These validations were essential to assess the robustness and generalizability of the identified evolutionary patterns. By evaluating the performance of ASCETIC on independent datasets, we ensured that the extracted evolutionary signatures consistently held predictive power across various cancer cohorts. We demonstrate the reliability and applicability of ASCETIC as a tool for uncovering consistent evolutionary patterns in cancer. Despite not being conclusive, this work paves the way for the definition of a curated catalogue of evolutionary signatures, along the lines of the widely used COSMIC Mutational Signatures database (https://cancer.sanger.ac.uk/signatures/).

## Results

### ASCETIC (Agony-baSed Cancer EvoluTion InferenCe)

The ASCETIC framework is based on the observation that, in most cases, the accumulation of passenger mutations during cancer progression follows random dynamics. However, a small set of mutations in driver genes are responsible for driving tumor evolution, and for these alterations, drift-driven evolution and selective pressures may lead to a consistent ordering across multiple patients[6].

Such ordering of driver mutations during cancer evolution may not be unique and can be confounded by heterogeneous cancer subtypes within a tumor dataset[7]. Therefore, ASCETIC decomposes the inference problem into three main tasks. First, it leverages the evolutionary models obtained during the pre-processing phase (Fig. 1) to build an agony-derived ranking[8] of the considered driver alterations, which portrays a *partial* temporal ordering among them (Fig. 2a). Second, our framework adopts a likelihood-based approach grounded in probabilistic causation theory[9] to perform model selection, returning a unique Bayesian Network that recapitulates the repeated evolutionary trajectories for that cancer (sub)type, i.e., the favored orderings among driver mutations (Fig. 2b). Third, ASCETIC uses the mutation co-occurrence patterns identified by such trajectories as features of a regularized Cox regression on survival data. This step allows one to cluster samples into different risk groups, whose significance is finally tested via standard Kaplan–Meier estimate. Overall, this model-informed feature selection allows ASCETIC to identify mutation (co-)occurrence patterns – the evolutionary signatures – exhibiting prognostic significance for any given cancer (sub)type.

ASCETIC produces two primary outputs: (1) a model that captures the recurring evolutionary trajectories observed across different patients during tumor evolution, and (2) a collection of genomic features (i.e., sets of single and co-occurring mutations) known as *evolutionary signatures*, which exhibits significant associations with clinical outcomes. ASCETIC provides the inferred risk levels for each selected evolutionary step and displays their relative prevalence within the cluster (Fig. 2c). Furthermore, ASCETIC also provides an estimate of uncertainty in the identified evolutionary steps through cross-validation, which enables us to pinpoint the most confident repeated trajectories among genes.

**Performance assessment via simulations.** The ASCETIC computational workflow comprises two primary steps. In the first step, ASCETIC performs the inference of a cancer evolution model, unveiling recurring evolutionary trajectories consistently observed across patients. The second step involves associating these inferred trajectories with

survival data, enabling the prediction of evolutionary steps that hold potential clinical significance. Referred to as evolutionary signatures, these signatures can be leveraged to stratify patients. Notably, ASCETIC's second step, which sets it apart from competing algorithms in the inference of cancer evolution models, represents a unique feature of the framework. Consequently, our simulations primarily focused on evaluating the performance of ASCETIC's first step in comparison to competing methods. To evaluate the performance of ASCETIC against competing methods, we conducted extensive tests on synthetic datasets, using two different noise models to generate the data. The first model mimicked samples obtained from NGS sequencing data, comprising a single biopsy per tumor, while the second model used single-cell or bulk sequencing NGS data to provide multiple samples for the same tumor (see Methods). We generated 8500 synthetic cancer evolution models (i.e., topologies) for a total of 26,500 different configurations and compared the performance of ASCETIC with that of the CAPRI algorithm[10] and the standard maximum likelihood fit approach for structure learning, in terms of accuracy, precision, recall (sensitivity), and specificity (see Methods and Supplementary Materials for details). Supplementary Fig. 2 shows the results comparing the accuracy of the different methods for a set of representative scenarios. Full results of the simulations are provided in the Supplementary Materials (see Supplementary Figs. 3–15).

Our results demonstrate that ASCETIC consistently outperforms the competing methods in all settings, presenting a very stable performance. Furthermore, these results underscore the substantial superiority of ASCETIC's expressivity compared to CAPRI. While CAPRI and other existing methods for analyzing repeated cancer evolution models are limited to inferring only conjunctive relations among genomic events[10], ASCETIC surpasses this constraint by providing partial orderings among genes and accommodating any type of temporal relation. This fundamental characteristic is evident in the notable decline in performance observed in the comparative methods relative to ASCETIC in the more general scenarios, where general temporal patters are simulated. This represents a crucial feature of our approach, empowering ASCETIC to deliver general, precise, and reliable models of cancer evolution.

**Performance assessment on cancer data.** We evaluated ASCETIC's performance on bulk cancer datasets of gliomas from three distinct studies: GLASS (222 patients[11]), MSK (924 patients[12]), and TCGA (1,122 patients[13]). Each patient's dataset included a single biopsy and copy number information. Gliomas can be categorized into three distinct molecular subtypes, which have been well-characterized in terms of gene mutations and copy-number alterations[14]. The first subtype is named G-CIMP (Glioma CpG island methylator phenotype), characterized by mutations in genes such as *IDH1/2*, *TP53*, and *ATRX*. The second subtype is named IDH mutant-codel, characterized by mutations in genes such as *IDH1/2*, *CIC*, and *FUBP1*, along with a chromosomal codeletion of 1p/19q. Finally, the third subtype is named *IDH1/2* wild-type, characterized by mutations in genes such as *TP53*, *EGFR*, and *PTEN* and various copy number alterations, but no alterations in *IDH1/2* genes. G-CIMP and *IDH1/2* mutant-codel subtypes are commonly observed in lower-grade gliomas and are associated with a favorable prognosis. On the other hand, *IDH1/2* wild-type subtypes are more common in glioblastomas and are associated with poor outcome. Based on this known ground truth, our objective was to evaluate the performance and reproducibility of ASCETIC by executing the entire framework independently on these three datasets. Notably, ASCETIC produced very consistent results (refer to Supplementary Figs. 89–98) and successfully identified three SNV evolutionary signatures along with their associated subtypes. These identified subtypes closely resembled the known subtypes of gliomas, exhibiting characteristic features and prognostic outcomes that matched the expected patterns.

To further assess the stability of ASCETIC's stratification feature, we performed a quantitative evaluation by independently repeating the framework on the GLASS dataset 1000 times. Specifically, we focused on the three clusters obtained from this dataset and generated synthetic data accordingly. Survival data was randomly sampled from the three GLASS subtypes, and molecular features were associated with them using empirically calculated cluster-specific distributions. Next, we applied ASCETIC to these 1000 simulated datasets, associating molecular and evolutionary features to prognosis, and measured the stability of the results by computing the Adjusted Rand Index (ARI) compared to the original inference. ASCETIC demonstrated a consistently good ARI, averaging above 0.75 (see Supplementary Fig. 92).

## Myeloid malignancies

To validate ASCETIC on experimental data, we applied it to a single-cell dataset obtained with the Tapestri Sequencing Platform for a set of myeloid disorders[15]. The dataset comprises single-cell mutational profiling of 146 samples from 123 distinct patients, including acute myeloid leukemia (AML) along with other myeloid malignancies such as clonal hematopoiesis and myeloproliferative neoplasms (MPN) (see Methods). Full results of the application of ASCETIC to these data are provided as Supplementary Data 1. In Supplementary Material Section 5 (top panel), we report and further discuss the inferred relations with a cross-validation score higher than 0.50.

Our analysis highlights that ASCETIC consistently identifies alterations in *CALR*, *JAK2*, and *IDH1/2* genes as early events in tumor history, while *NRAS* represents an acquired secondary event, towards which evolutionary trajectories appear to converge during the progression of the disease. These observations are consistent with published data and support the utility of ASCETIC in predicting evolutionary steps in myeloid malignancies[16,17]. The co-occurrence of *CALR* and *ASXL1* mutations has been reported to be enriched in patients affected by essential thrombocythemia (ET). In particular, it has been shown that an additional *ASXL1* mutation in *CALR*-mutated patients worsen the *CALR* phenotype[18,19].

Furthermore, ASCETIC identifies a frequent *NPM1*-to-*FLT3* evolution with high cross validation score (Supplementary Material Section 5, top panel). This is also in line with clinical observations in AML, where the presence of mutated *NPM1* in the absence of a *FLT3* mutation identifies a subset of AML with favorable prognosis, while the subsequent acquisition of a *FLT3* mutation confers an intermediate risk[20]. ASCETIC also correctly positions *DNMT3A* as a parent mutation with very high confidence[21,22]. Finally, single-cell analysis of myeloid malignancies performed by Miles and colleagues[15] revealed the co-occurrence of *IDH1/2* and *TET2* mutations, which had never been reported before in myeloid neoplasms, as the two mutations were previously described to be mutually exclusive[23]. ASCETIC not only detects their co-occurrence but also orders their accumulation over time, suggesting that *IDH1/2* variants are acquired before *TET2* mutations.

These results demonstrate that ASCETIC can reliably predict mutational trajectories from single-cell data and build models of disease evolution for myeloid malignancies.

**Acute myeloid leukemia (Fig. 3) — 4 SNV evolutionary signatures.** Unfortunately, survival data were not available for the single-cell dataset by Miles and colleagues[15]. However, ASCETIC can integrate sequencing experiments at varying resolutions. Therefore, to demonstrate the versatility of ASCETIC and validate its findings, we utilized the evolutionary model of myeloid disease obtained from single-cell data to stratify an external dataset comprising 366 tumor samples collected from patients affected by AML[24]. The evolutionary model was obtained by applying ASCETIC to the single-cell dataset of myeloid malignancies described above[15]. Samples were stratified into

different risk groups by selecting the alterations associated with the minimum cross-validation error. Stratification was performed by ASCETIC and then survival analysis comparing the different groups was carried out via standard Kaplan–Meier estimates (see Methods). This analysis resulted in the extraction of four SNV evolutionary signatures identifying four clusters that exhibit significantly different survival rates ($p < 0.001$).

As shown in Fig. 3, the three most prevalent evolutionary routes characterizing AML SNV Evolutionary Signature (AML SNV Evo Sig) #1 involve alterations in *IDH1/2*, *RAD21*, *DNMT3A* and *KIT* genes as early evolutionary steps, with later acquisition of *TET2* alterations (Fig. 3 and Supplementary Fig. 16). Overall, this signature cumulates several steps with favorable clinical projection. Indeed, cluster 1, associated to AML SNV EvoSig#1, shows the best overall survival (OS). In this cluster, evolution from *IDH1/2* to *TET2* appears to be relevant for survival, conferring a lower risk of progression, possibly defining a subgroup of patients with hypermethylated phenotype. When *TET2*-mutated AML patients where partitioned into *TET2 vs IDH1/2*-to-*TET2* subgroups, the latter had a significantly better survival, corroborating our findings (Fig. 3 and Supplementary Data 2). The prognostic significance of *RAD21* mutations in AML is controversial as some studies described them as independent factors for a longer OS, while others found no differences in survival[25]. According to ASCETIC, mutated *RAD21* is associated with better outcome, shedding new light on the prognostic value of *RAD21* mutations in AML.

AML SNV EvoSig #2 is also characterized by mutations of *IDH1/2* genes, which are then followed by alterations in *DNMT3A*, rather than *TET2*, thus defining a group of patients with intermediate risk. A fraction of these patients further acquires *NRAS* mutations. These results suggest that alternative evolution routes (*DNMT3A vs TET2*) from a common first hit (*IDH1/2*) may change AML prognosis. AML SNV EvoSig #3 is defined by a clear trajectory involving evolution from *NPM1* to *FLT3* and is associated with poor outcome. This is a well-documented risk factor in AML, as the presence of mutations in *FLT3* is often associated with reduced OS, particularly in *NPM1*-mutated patients, and improves risk stratification in patients without cytogenetic abnormalities[26]. Remarkably, both AML SNV EvoSig #1 and AML SNV EvoSig #3 exhibit mutations in the *NPM1* gene, which displays a significant association with clinical outcomes. However, in AML SNV EvoSig #3, additional mutations in the *FLT3* gene are observed, and they are notably correlated with survival. ASCETIC's inference reveals that the evolutionary step from *NPM1* to *FLT3* is also linked to a (poor) prognosis. These two signatures, represent recurring cancer evolutions that share common elements, but follow distinct progressions. In addition, several patients carrying mutations of the *STAG2* cohesin are found in cluster 3, confirming previous findings by Papaemmanuil and colleagues, who identified a chromatin/spliceosome group, including *STAG2* variants, with overall poor prognosis[27]. Finally, evolutionary trajectories involving alterations in *TP53* define SNV EvoSig #4, and patients belonging to cluster 4 show the shortest median OS (< 10 months). As a matter of fact, according to the latest International Consensus Classification of myeloid neoplasms, *TP53*-mutated AML should be considered as an independent entity associated with a lower likelihood of response to conventional chemotherapy and poor outcome and should be included in the adverse prognostic risk category[28]. In addition, ASCETIC discovers that evolutionary routes involving *PHF6* (11% *vs* 1%; odds-ratio (OR) = 16.2; $\chi$-squared $p < 0.001$) and *U2AF1* (23% *vs* 0%; OR = 61.3; $\chi$-squared $p < 0.001$) are also common in cluster 3 and are significantly enriched with respect to the other two clusters. Mutations occurring in *PHF6* have been described in AML and showed an unclear association to disease progression[29], while *U2AF1* alterations have been correlated with poor prognosis in AML[30], thus corroborating our findings.

**Stratification based solely on mutations.** To demonstrate the significance of ASCETIC results, we conducted an analysis based solely on somatic mutations. When ignoring evolutionary arcs, some

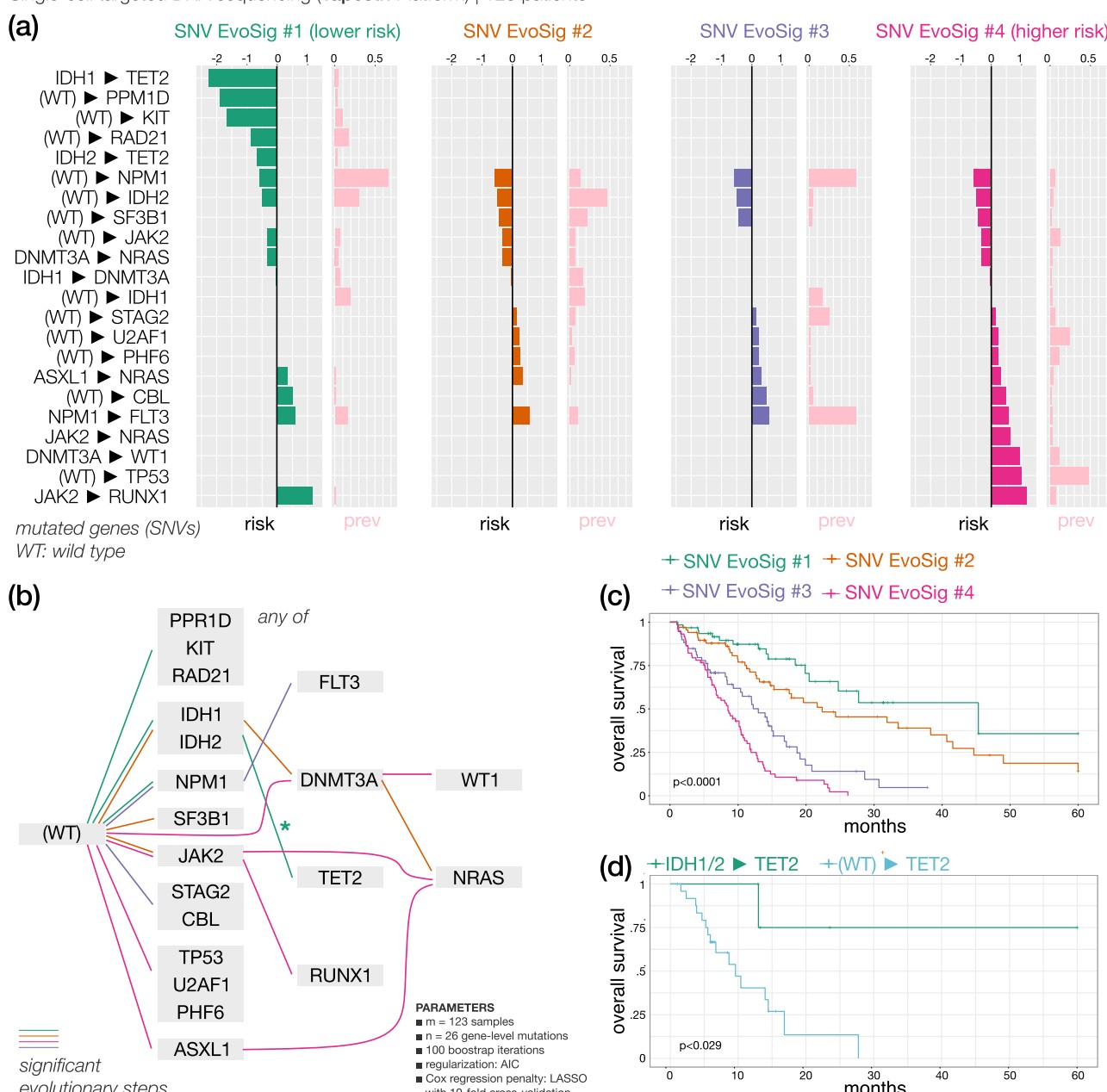

**Fig. 3 | ASCETIC analysis of acute myeloid leukemia samples**[24] **(123 patients).** We show the 4 extracted SNV evolutionary signatures (**a**), the evolutionary model of repeated evolution inferred by the framework (**b**) and the Kaplan–Meier analysis (log-rank *p*-value) of the associated evolutionary signatures (**c**, **d**).

information was lost. Supplementary Figs. 17–18 provide examples of the reduced resolution, such as failing to distinguish high-risk *NPM1*-to-*FLT3* patients from low-risk *FLT3* wild-type patients. In addition, mutation analysis without evolution fails to split *NRAS* mutated patients into intermediate and poor outcome groups, as it lacks information on the parent event occurring before the acquisition of *NRAS* mutations.

Collectively, these analyses conclusively demonstrate that ASCE-TIC effectively identifies subgroups of AML patients with distinct survival outcomes, surpassing the capabilities of conventional mutation-based patient clustering methods. Notably, while ASCETIC can capture straightforward associations to prognosis as detected by single-gene mutation analyses, it goes beyond that by detecting additional prognostic associations based on evolutionary patterns.

**Validation of ASCETIC's evolutionary model on unseen data.** We validated ASCETIC's evolutionary steps on acute myeloid leukemia samples using unseen single-cell data from 123 AML patients[31]. We evaluated the mutational trees reported in the original work by Morita and colleagues[31] for each evolutionary step inferred by ASCETIC, checking if they were consistent with the model by ASCETIC or if they presented discrepancies. We report this analysis in Supplementary Material Section 6. We examined the 17 evolutionary steps inferred by ASCETIC and provided the following information: (1) the number of phylogenies/patients in which the same evolutionary step was observed, (2) the number of phylogenies/patients in which an evolutionary step inconsistent with ASCETIC's inference was observed, and (3) the number of phylogenies/patients in which no supporting evidence for the evolutionary step was identified. Notably, most of the

evolutionary steps (15 out of 17) returned by ASCETIC were highly consistent with the mutational trees. Therefore, our results provide strong and direct validation of ASCETIC's model of acute myeloid leukemia evolution.

**Validation of ASCETIC's AML SNV evolutionary signatures on unseen data.** We validated ASCETIC's AML SNV evolutionary signatures on Pan-Cancer Atlas[32] unseen data. Specifically, the Pan-Cancer Atlas AML dataset comprises data from 200 patients, predominantly collected at the time of diagnosis. Despite ASCETIC's model being constructed based on advanced cancers, the results demonstrated remarkable consistency, with ASCETIC successfully identifying three out of four subtypes in the validation dataset (Supplementary Figs. 99–100). Furthermore, our results demonstrate that these signatures can effectively stratify patients into groups with significantly different prognoses ($p < 0.0001$). Specifically, we have identified a group of patients carrying *NPM1* and *FLT3* mutations (44 patients), which exhibit a poor prognosis (< 25% survival rate at 24 months), and a cluster of patients with *TP53* mutations (13 patients), who have the lowest survival, as observed in the original dataset (all patients died within 18 months).

Finally, we highlight ASCETIC's model-informed feature selection capabilities. Applying LASSO Cox regression with many predictors (e.g., testing all genes co-occurrence) has known pitfalls such over-regularization, unstable variable selection, and high computational load. Collinear predictors can lead to the loss of valuable information, and irrelevant ones add noise[33,34]. Our approach mitigates these limitations.

## Non-small cell lung cancer

Lung cancer is a leading cause of cancer-related deaths worldwide and encompasses a diverse group of pathologies that are classified into two main categories based on morphology, immunohistochemistry, and molecular features: small cell and non-small cell lung cancer (NSCLC)[35]. Despite significant progress in lung cancer treatments, the 5-year overall survival (OS) rate remains very low (15%).

**Early-stage NSCLC (Fig. 4) — 3 SNV evolutionary signatures.** We analyzed the data from the TRACERx research project[36] comprising multi-region sequencing samples for a total of 302 biopsies from 100 early-stage NSCLC patients, and a total of 65,421 somatic substitutions (see Methods). Full results of ASCETIC applied to these data are provided as Supplementary Data 3. We report in Supplementary Material Section 5 (bottom panel) and further discuss here the inferred relations with a cross validation score higher than 0.50.

Several mutations that have been classified as cancer drivers in NSCLC were identified by ASCETIC as early alterations (see Supplementary Material Section 5, bottom panel) involved in the evolution of this cancer type: among them *TP53*, *KRAS*, *KEAP1*, *CDKN2A*, *PIK3CA*, *ATM*, *EGFR*, *BRAF*, *KMT2D*, *EP300*, and *FBXW7*[37]. On the other hand, *ARID1B*, *EP300*, *NFE2L2*, and *PTPRC* mutations emerge as nodes of convergence for different evolutionary routes. ASCETIC confirmed the mutual exclusivity of some driver mutations, such as *EGFR* and *KRAS*[38] or *TP53* and *KEAP1*[39], clarified the timing relation between genomics alterations that are involved in lung carcinogenesis and allowed us to discover new late-occurring mutations that may have a relevant role in NSCLC progression.

Survival data associated with this cohort allowed the stratification of patients and identification of SNV evolutionary signatures (Fig. 4), although the observed evolution is rather limited in the early-stage setting. We found that early acquisition of *MGA*, *FGFR1*, and *ATM* mutations confer favorable prognosis, while alterations of *WT1* and cell cycle regulators (*CCND1* and *CDKN2A*) yield a signature of negative prognosis. Interestingly, an evolution from *CDKN2A* to *CYLD* depicts a particularly unfavorable outcome compared to patients lacking

mutation of *CYLD* (Fig. 4, Supplementary Data 2). The *CYLD* tumor suppressor gene encodes for a deubiquitinating enzyme involved in cylindromatosis syndrome, characterized by multiple skin benign tumors. Somatic *CYLD* mutations have been found in several cancer types[40].

**Metastatic lung adenocarcinoma (Fig. 5) — 3 SNV evolutionary signatures.** Since localized NSCLC data provided limited evolutionary information, we also analyzed advanced NSCLC data using ASCETIC. Specifically, we exploited the MSK-MET dataset[41], which includes both genomic and clinical information on metastatic NSCLC. Our analysis first focused on lung adenocarcinoma (LUAD), which traditionally has been classified based on morphology into various subtypes, including in situ, minimally invasive, invasive non-mucinous, invasive mucinous, colloid, fetal, and enteric-type adenocarcinoma[35]. From a genetic perspective, frequent driver mutations in LUAD include *EGFR*, *KRAS*, *HER2*, *BRAF*, and *PIK3CA*. Analyzing a dataset of 1176 LUAD patients from MSK-MET using our algorithm, we identified three subtypes based on three distinct SNV evolutionary signatures.

All three groups are characterized by *KRAS*/*TP53* progression. The reported frequency of *KRAS*/*TP53* co-mutation ranges from 31% to 46%[42]. In our analysis, we observed that this feature is broadly conserved across different clusters, but it is not significantly associated with the outcome of a particular subtype. In contrast, the convergence of different routes onto *RBM10* mutations seems to provide better survival to patients showing LUAD SNV EvoSig #1, while the late acquisition of mutations in *SMARCA4* markedly increases risk for patients of cluster 3. Interestingly, clonal *RBM10* loss of function has been associated with shorter survival in early-stage lung cancer patients[43], while in our analysis subsequent evolution towards *RBM10* mutations predicts favorable prognosis in metastatic samples, indicating that the timing of mutation acquisition during tumor history is relevant. cluster 3 is also characterized by high prevalence of early *KEAP1* mutations (Supplementary Fig. 20), which anticipate in general a dismal outcome unless followed by *RBM10* mutations. Of note, the direction of tumor progression from an initial *KRAS* mutation appears to dictate subsequent prognosis: median OS is 12 months *vs* not reached in *KRAS* to *SMARCA4* and *KRAS* to *EPHA3* groups, respectively (Fig. 5). Interestingly, *SMARCA4* alterations have been associated with shorter OS in patients with metastatic NSCLC[44].

Additional internal subgrouping of patients according to specific trajectories revealed interesting survival differences, e.g., in patients with *KRAS* mutations that acquire or not *EPHA3* variants, patients with *EGFR* or *KEAP1* alterations evolving or not to mutant *RBM10*, and patients with *EGFR* only or *EGFR*-to-*NF1*/*PIK3CA* genes mutations (Fig. 5 and Supplementary Data 2). Finally, cluster 2 shows an intermediate survival and an associated LUAD SNV EvoSig #2 with no specific evolutionary pattern other than the common *KRAS*/*TP53* co-mutation. Likely, patients in this group carry additional genetic events that we did not consider in our analysis, such as copy number variations.

**Validation of ASCETIC's LUAD SNV evolutionary signatures on unseen data.** We validated ASCETIC's LUAD SNV evolutionary signatures on Pan-Cancer Atlas[32] unseen data. The LUAD dataset within the Pan-Cancer Atlas consists of information gathered from 566 patients, primarily at the time of their diagnosis. Also in this case, ASCETIC's model was developed specifically for metastatic lung adenocarcinomas. Nevertheless, the findings showed impressive uniformity, as ASCETIC accurately detected all three subtypes of LUAD in the validation dataset (Supplementary Figs. 101–102).

We further validated the prognostic potential of the discovered clusters. ASCETIC identified three evolutionary signatures of lung adenocarcinoma. These signatures successfully stratified the patients in the validation cohort into three distinct groups (48, 200, and 10 patients, respectively), exhibiting significant differences in survival ($p < 0.0001$). The observed clusters consistently demonstrated similar

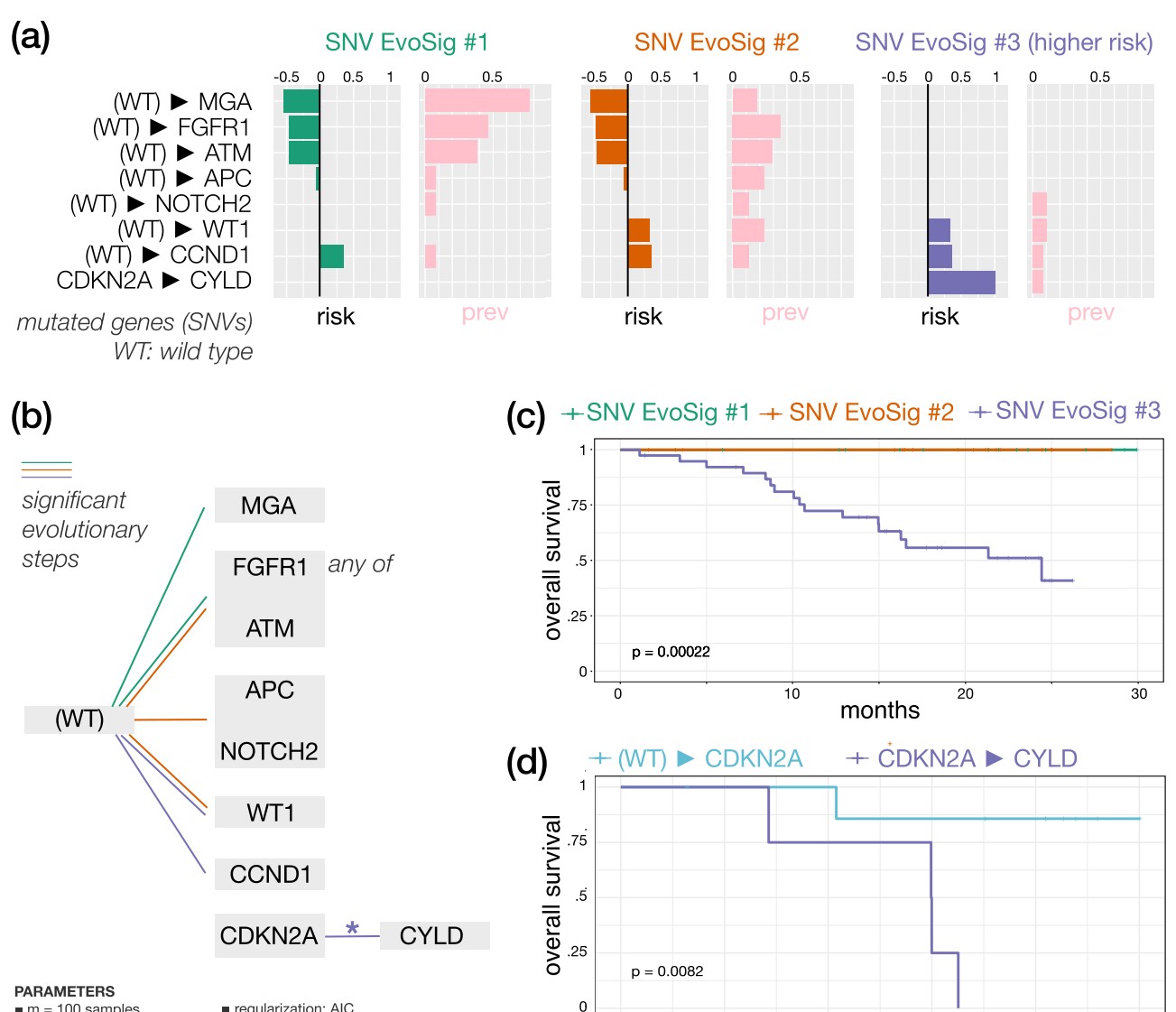

# SNV Evolutionary signatures of Non-Small Cell Lung Cancer (NSCLC)
## Multi-region whole-exome sequencing (TRACERx) | 100 patients

**Fig. 4 | ASCETIC analysis of early-stage non-small cell lung cancer samples from the TRACERx research project[36] (100 patients).** We show the 3 extracted SNV evolutionary signatures (**a**), the evolutionary model of repeated evolution inferred by the framework (**b**) and the Kaplan–Meier analysis (log-rank p-value) of the associated the evolutionary signatures (**c**, **d**).

behaviors in the validation datasets, with patients of clusters 1 and 2 showing a survival rate at 32 months of 75% and 50%, respectively, in both testing and validation cohorts, while patients carrying evolutionary trajectories toward *SMARCA4* (cluster 3) exhibit the worst survival outcomes with a 35% survival rate at 32 months.

**Validation on external data of the three LUAD subtypes.** We validated the existence of 3 subtypes discovered by ASCETIC for lung adenocarcinoma on unseen bulk data for more than 400 patients from three different datasets[45–47]. To this end, we built a classifier by standard random forest (mean of squared residuals 0.021; explained variance 90.53) to predict the subtype classification obtained by ASCETIC on LUAD samples and used such classifier to perform predictions for the patients in the validation dataset. We report oncoprint and survival analysis performed on the LUAD dataset (Supplementary Figs. 21–22), whose subtypes show very consistent

mutational profiles and significant differences in OS (p = 0.0054) as predicted by ASCETIC.

**Metastatic lung squamous cell carcinoma (Fig. 6) — 3 SNV evolutionary signatures.** Squamous cell carcinoma (LSCC) represents approximately 30% of all NSCLC cases. Standard treatment is still based on chemotherapy doublets. Traditionally, four histologic subtypes are identified: clear cell, small cell, papillary, and basaloid, the latter being an aggressive form with shortest survival. Attempts to molecularly define this cancer led to the identification of *FGFR1, PI3K, TP53,* and *DDR2* genes as possible therapeutic targets, but this has not been translated into effective treatments thus far. More recently, four LSCC subgroups (primitive, classical, secretory, and basal) were identified by gene expression data across different datasets, with the primitive type showing significantly worse survival[48,49]. Several other studies clustered patients by gene expression/methylation,

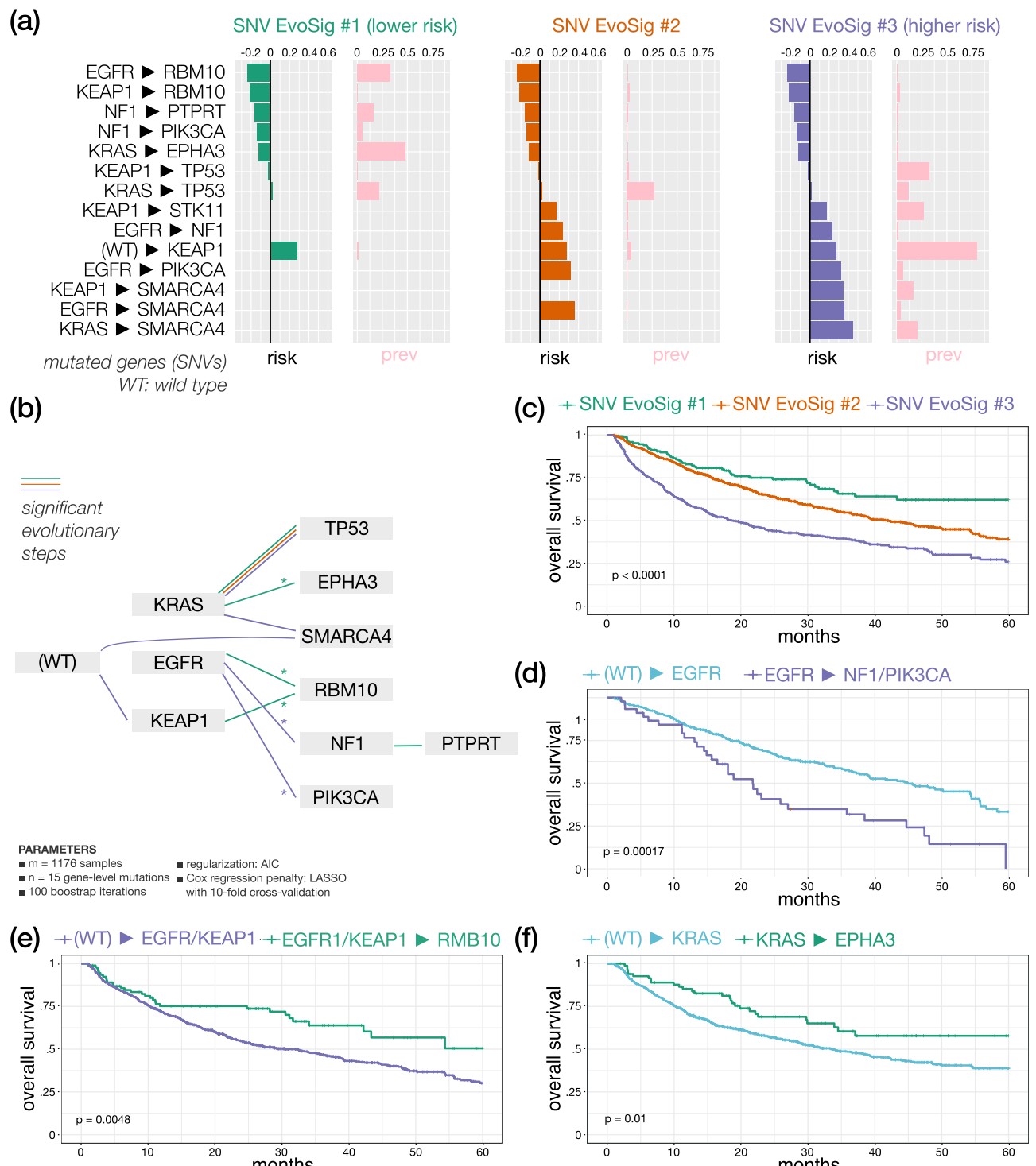

**Fig. 5 | ASCETIC analysis of lung adenocarcinoma metastatic samples from MSKMET[41] (1176 patients).** We show the 3 extracted SNV evolutionary signatures (**a**), the evolutionary model of repeated evolution inferred by the framework (**b**) and the Kaplan–Meier analysis (log-rank *p*-value) of the associated the evolutionary signatures (**c**–**f**).

while no such classification has been attempted by mutation over-representation or phylogenies.

We analyzed mutational data of 202 metastatic LSCCs from the MSK-MET dataset: the three SNV evolutionary signatures identified by ASCETIC classified patients into three risk groups with significantly different survival, based on the relative frequency and ordering of somatic alterations. Overall, the most prevalent alterations involved genes such as *TP53*, *CDKN2A*, *KEAP1*, and *KMT2D*. These genes were among the 10 significantly mutated genes previously identified by the TCGA data analysis[49].

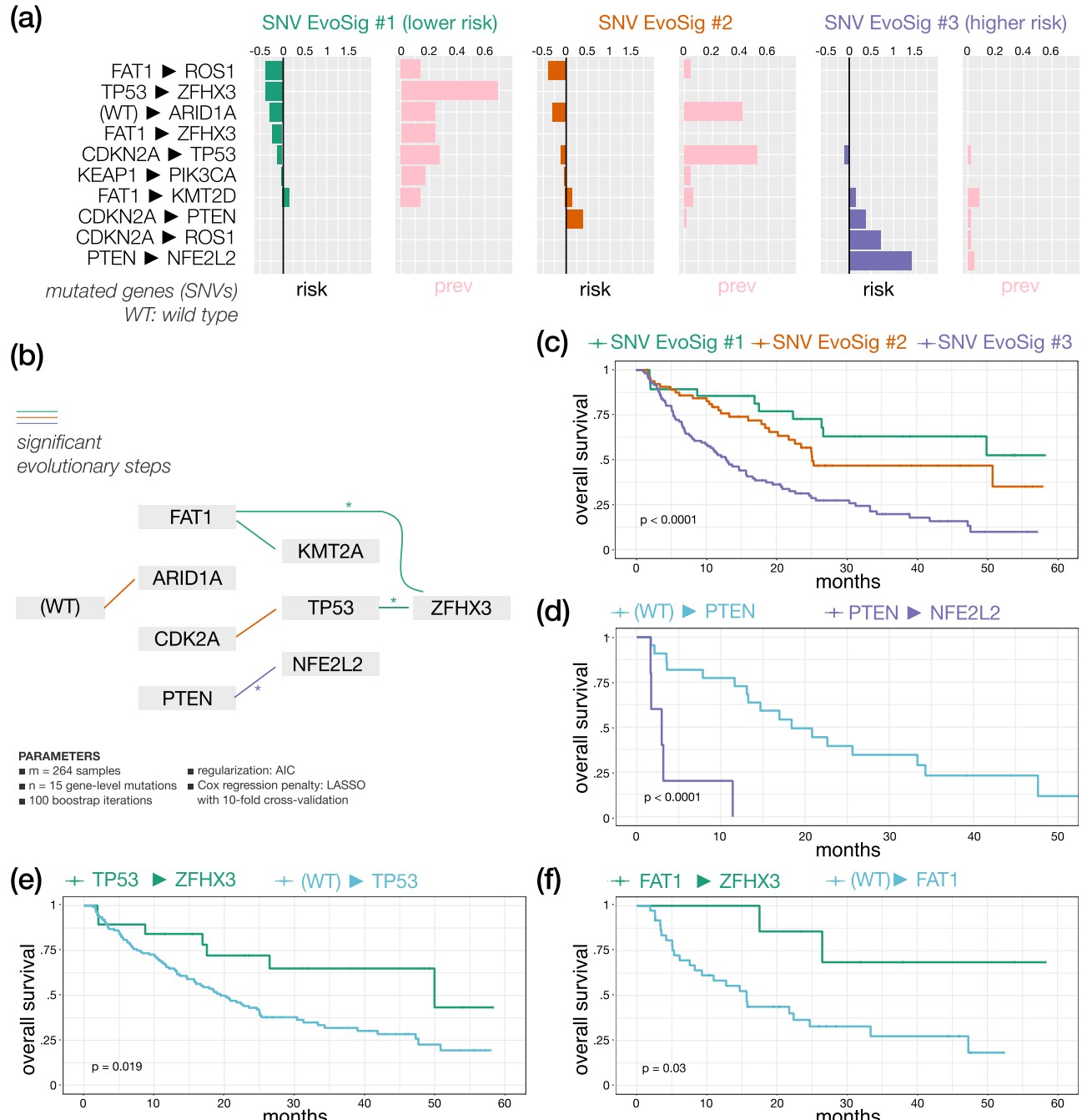

**Fig. 6 | ASCETIC analysis of lung squamous cell carcinoma metastatic samples from MSK-MET[41] (264 patients).** We show the 3 extracted SNV evolutionary signatures (**a**), the evolutionary model of repeated evolution inferred by the framework (**b**) and the Kaplan–Meier analysis (log-rank *p*-value) of the associated the evolutionary signatures (**c**–**f**).

The best outcome was associated to LSCC SNV EvoSig #1 involving mutations in *FAT1* and *TP53* with convergent evolution toward the *ZFHX3* (Zinc finger homeobox 3, also known as *ATBF1*) gene. *ZFHX3* mutations are highly over-represented in cluster 1 compared to clusters 2 and 3 (72% *vs* 2%; OR = 73.1; χ-squared *p* < 0.0001; Supplementary Fig. 23) and invariably occur as a later event. Stratification of patients according to the acquisition of *ZFHX3* mutations after *TP53* or *FAT1* versus all other *TP53*- or *FAT1*-mutated cases showed a marked difference in OS. *ZFHX3* is a putative tumor suppressor gene and has been frequently found mutated in solid cancers[50]. Recent work identified *ZFHX3* mutations as an independent prognostic biomarker of longer OS in NSCLC patients treated with immune checkpoint inhibitors[51,52], possibly explained by higher mutational burden and increased tumor immunogenicity[52]. In contrast, LSCC SNV EvoSig #3 is associated with the worst outcome and is characterized by a *CDKN2A*, *PTEN* and *NFE2L2* progression, while evolution from *CDKN2A* to *TP53* describes LSCC SNV EvoSig #2, and is linked to a moderate risk. Within cluster 3, *PTEN* mutated patients showed a strikingly short OS when

progressing to *NFE2L2* (see Fig. 6 and Supplementary Data 2, median OS 3 vs 18 months in *PTEN*-to-NFE2L2 *vs* PTEN cases, respectively; *p* < 0.0001).

**Validation of ASCETIC's LSCC SNV evolutionary signatures on unseen data.** We validated ASCETIC's LSCC SNV evolutionary signatures on Pan-Cancer Atlas[32] unseen data. The LSCC dataset within the Pan-Cancer Atlas includes information obtained from 487 patients, primarily gathered during the diagnosis stage. Although ASCETIC's model being specifically developed for metastatic lung squamous cell carcinomas, the outcomes exhibited remarkable consistency. ASCETIC effectively identified all three subtypes of LSCC in the validation dataset, showcasing its good performance (Supplementary Figs. 103–104). We conducted additional validation to assess the prognostic potential of the identified clusters. ASCETIC detected three distinct evolutionary signatures in lung squamous cell carcinoma. These signatures effectively categorized the patients in the validation cohort into three groups (47, 62, and 126 patients, respectively), revealing notable variations in survival rates (*p* = 0.009). The observed clusters consistently displayed analogous behaviors in the validation datasets. In particular, the patients carrying co-occurrent mutations in *TP53* and *ZFHX3* genes showed the best prognosis, with a survival rate after 1 year above 80%.

### Large-scale sequencing and survival data analysis
We applied ASCETIC to the Pan-Cancer Atlas[32] and the MSK-MET[41] datasets, considering a total of more than 35,000 samples across most cancer types (see Methods). All the evolutionary models returned by ASCETIC on these datasets are provided as Supplementary Data 4 and 5.

After performing the ASCETIC framework on these data, we assessed the presence of significant differences in prognosis correlated to the evolution models returned by our framework. To this end, we selected for each cancer type the set of alterations associated with the minimum cross-validation error and stratified the patients into different risk groups. We then performed survival analysis of the different risk groups via standard Kaplan–Meier estimate (see Methods). We uncovered different SNV evolutionary signatures from a total of 25 cancer (sub)types showing significant differences in OS.

Furthermore, a comprehensive pan-cancer analysis was conducted by aggregating the results obtained from the inferences performed in the individual subtype. This approach yielded insights into the recurrent evolutionary trajectories observed across different cancer types. By combining the results, we gained a deeper understanding of the shared patterns and trends of evolution at the pan-cancer level. These results are reported in Supplementary Data 6.

We discuss in detail the results for 2 selected cancer types, leaving the remaining analyses as Supplementary Materials (see Supplementary Figs. 26–88).

**Prostate cancer (Fig. 7) — 3 SNV evolutionary signatures.** Prostate cancer (PCa) is the second most common tumor in men worldwide. Metastatic PCa samples (*n* = 280) were clustered by ASCETIC in three SNV EvoSig subgroups (Fig. 7). While all three carry *TP53* mutations, typical of advanced PCa[53], subsequent alterations in *KMT2C, AR,* or *CTNNB1* appear to confer significantly poor outcome, while evolution to *GRIN2A* grants a more favorable of disease. Of note, *KMT2C*-mutated PCa patients have been previously shown to have a reduced disease-free survival[54]. Furthermore, *TP53, AR,* and *KMT2C* were the top three differentially altered genes in a cohort of 150 castration resistant PCa cases compared to primary tumors[55]. Similarly, cell-free DNA sequencing revealed activating *CTNNB1* mutations in patients with castration-resistant disease at progression[56]. Thus, ASCETIC recapitulated all most unfavorable events of PCa evolution in SNV EvoSig #3. In contrast, cluster 1, displaying the longest OS, shows a collection of low-risk evolutionary arcs, including an interesting *KMT2D, ZFHX3,* and *MED12* progression. Thus, late acquisition of *ZFHX3* mutations is associated

with better prognosis in at least two different cancers, i.e., NSCLC and PCa. Interestingly, PCa-specific *MED12* mutations were shown to reduce the assembly of the Mediator complex, an important regulator of PCa progression to castration resistance[57]. In a recent analysis, mutations of *KMT2D* and *MED12* were found over-represented in the group of patients with good prognosis[58]. Lastly, an intermediate survival is associated with SNV EvoSig #2, which partially overlaps with SNV EvoSig #1, but without further evolution to low-risk variants like *GRIN2A* (Fig. 7, Supplementary Data 2).

**Validation of ASCETIC's PCa SNV evolutionary signatures on unseen data.** We validated ASCETIC's PCa SNV evolutionary signatures on two external datasets, the first obtained from the Pan-Cancer Atlas[32] and the second from the SU2C study[59]. The Pan-Cancer Atlas PCa dataset provides data from 494 primary patients. ASCETIC's model, constructed on metastatic prostate cancer, was still capable of successfully identifying two out of three PAc subtypes (156 and 58 patients, respectively) in this first validation dataset (Supplementary Figs. 105–106). We further conducted an additional validation using the dataset from SU2C[59], which encompasses data from 444 cases of advanced prostate cancers. In this second validation, ASCETIC successfully identified and retrieved all the three subtypes (6, 48, and 6 patients, respectively) inferred on the MSK-MET cohort (Supplementary Figs. 107–108).

In both validation datasets, ASCETIC successfully established a correlation between the identified evolutionary signatures and significant differences in survival (*p* = 0.022 in the first validation dataset and *p* = 0.00026 in the second). These discovered signatures consistently exhibited an association with prognosis across both validation datasets as well as the training dataset.

**Endometrial cancer copy number low subtype (Fig. 8) — 4 SNV evolutionary signatures.** Endometrial cancer (EC) is the fourth most common malignancy among women. It is commonly classified in two histo-pathological groups with different prognosis[60]. EC patients (with low copy number variation) were sorted by ASCETIC into 4 subgroups based on SNV evolutionary signatures (Fig. 8). All clusters share an evolutionary step through *ARID1A*. *ARID1A* is a tumor suppressor gene in EC: its loss was reported to have a role in EC initiation and progression[61]. However, *ARID1A* mutations do not seem to be associated with differential survival in EC patients in the considered copy number low subtype, according to our analysis. Similarly, alteration of *CHD4* gene is found in all clusters and thus, it is not prognostic. However, subsequent evolution from *CHD4* to *PIK3CA* mutations confers better prognosis to clusters 1 and 2. Mutations of *CHD4* and *PI3KCA* have been described to drive progression from hyperplasia to EC but have not yet been associated to prognosis[62]. In addition, clusters 1 and 2 (showing similar signatures and survival) share common early lesions in other genes such as *ARHGAP35*, *MED12*, *BCOR*, and *SOX17*. Cluster 3 is associated to SNV EvoSig #3, characterized by early mutations of *FGFR2* with secondary mutations in *ARID5B*. Interestingly, *FGFR2* activating mutations have been previously associated with shorter disease-free survival in stage I/II endometrioid EC patients[63]. Finally, cluster 4 shows the worst OS; it is associated to endometrial SNV EvoSig #4, defined by high prevalence of secondary alterations in *CTCF* gene, in a typical *PIK3CA*-to-*CTCF* evolution. In line with our analysis, *CTCF* mutations have been associated with EC relapse, metastasis, and poor survival[64]. To further cross validate this finding, we observed that patients with a *PIK3CA*-to-*CTCF* evolution have significantly shorter OS compared to all other *PIK3CA*-mutated patients who did not acquire *CTCF* mutations (Figs. 7 and 8).

## Discussion
The accumulation of alterations in certain driver mutations can follow repeated routes in different cancer patients. Therefore, detecting such trajectories could be crucial for implementing appropriate therapeutic

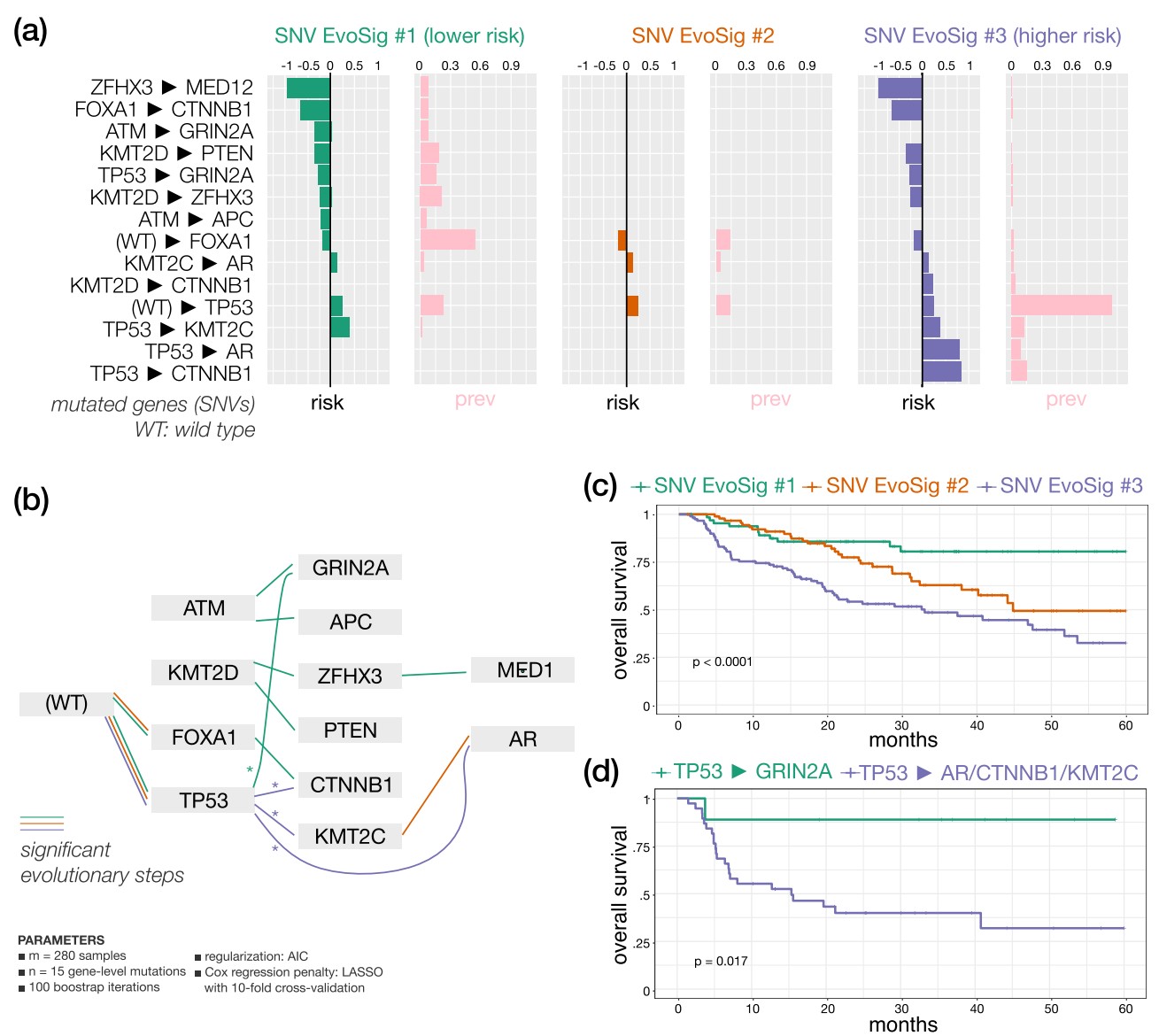

**Fig. 7 | ASCETIC analysis of prostate cancer samples from MSK-MET[41] (280 patients).** We show the 3 extracted SNV evolutionary signatures (**a**), the evolutionary model of repeated evolution inferred by the framework (**b**) and the Kaplan−Meier analysis (log-rank *p*-value) of the associated the evolutionary signatures (**c**, **d**).

responses. In fact, being able to stratify cancer patients based on their molecular evolution could enable the prediction of the future steps of the disease progression, potentially allowing the implementation of optimal and personalized treatments that anticipate the next stages of the cancer's evolution.

The ASCETIC framework enables the extraction of temporal relationships among driver alterations that show consistent accumulation dynamics across multiple patients during tumor progression. The method leverages model selection combined with an extended version of Suppes' probabilistic approach to causality and can process heterogeneous genomics datasets in an unsupervised manner. This approach allowed us to discover the existence of cancer (sub)type-specific *single nucleotide variants (SNV) evolutionary signatures* associated to different diseases outcomes. These signatures represent the "favored trajectories" of acquiring driver alterations during cancer evolution that can be used to stratify patients and characterize molecular evolution dynamics leading to potentially different prognostic responses.

Notably, the concept of SNV evolutionary signatures complements that of single base substitution (SBS) mutational signatures. While SBS mutational signatures model the mechanistic mutational processes that cause alterations in any cancer genome, SNV evolutionary signatures reveal conserved patterns of acquiring functionally advantageous single nucleotide variants that emerge during cancer development.

It is important to note that ASCETIC does not rely on pre-annotated driver mutations as input data. Instead, ASCETIC operates on the premise that the accumulation of passenger mutations during cancer progression may occur randomly among different patients. However, a small subset of driver mutations, responsible for driving tumor evolution, may exhibit consistent ordering across multiple patients. Therefore, ASCETIC identifies these small sets of genes that consistently appear in a specific order, which can be considered as driver mutations. By definition, the repeated evolutions inferred by ASCETIC involve these driver mutations, shedding light on their crucial role in cancer development and progression.

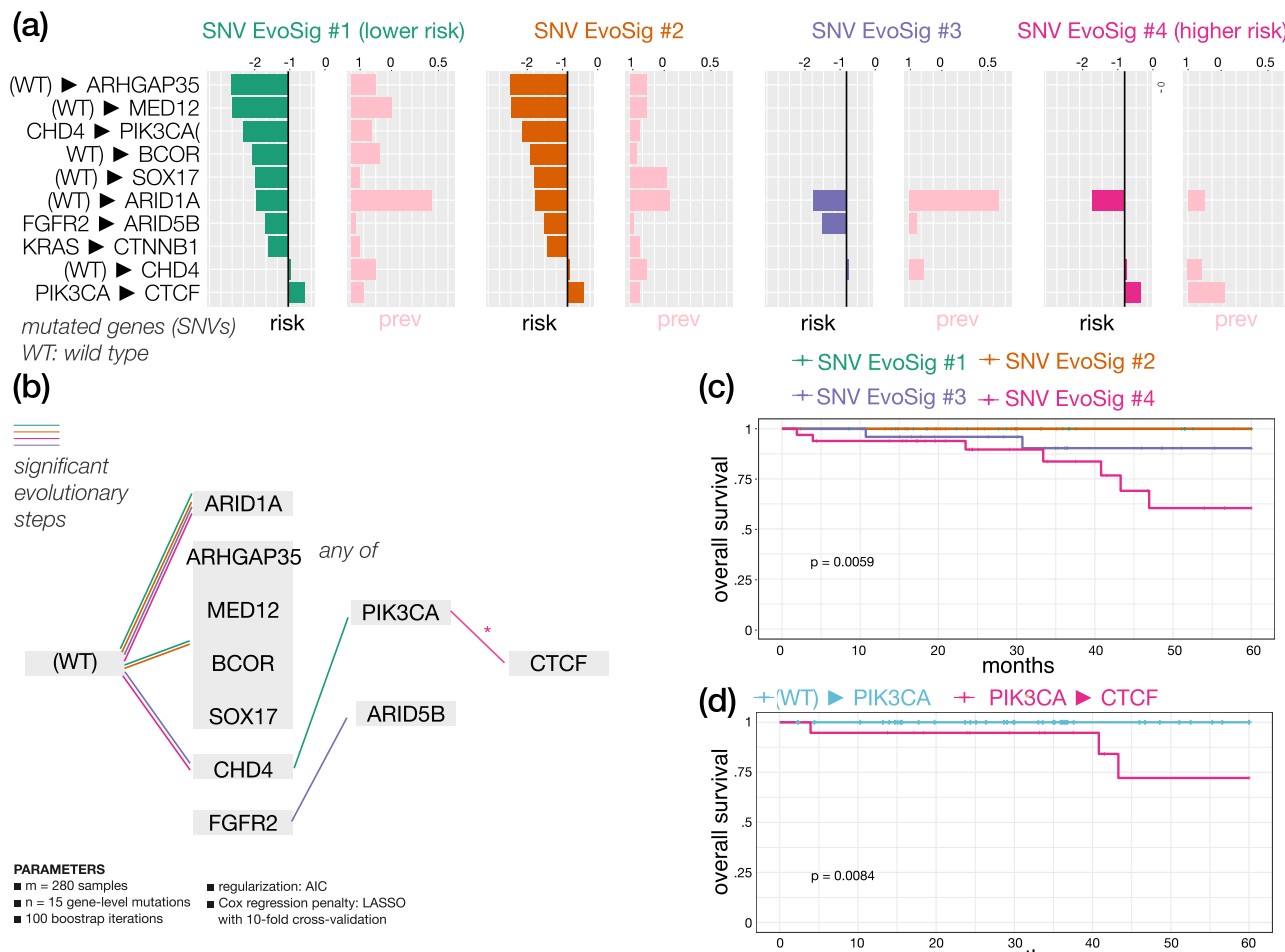

**Fig. 8 | ASCETIC analysis of endometrial cancer samples from Pan-Cancer Atlas[32] copy number low subtype (147 patients).** We show the 4 extracted SNV evolutionary signatures (**a**), the evolutionary model of repeated evolution inferred by the framework (**b**) and the Kaplan–Meier analysis (log-rank p-value) of the associated the evolutionary signatures (panels **c**–**d**).

The ASCETIC framework has been intentionally designed with modularity in mind. This unique characteristic allows for the integration of various cutting-edge phylogenetic tools to infer mutational trees, which serve as input to ASCETIC. Moreover, our method incorporates a bootstrap and resampling scheme, enhancing the statistical robustness of the findings. While all computational methods for cancer data analysis can be influenced by sampling bias or issues related to noise and resolution, our tool employs advanced statistical approaches to address these challenges. Furthermore, ASCETIC provides a cross-validation score for each generated evolutionary trajectory. This feature ensures a comprehensive evaluation of its reliability and offers a direct estimation of the uncertainty associated with the results.

We demonstrated the effectiveness of our method by analyzing cancer sequencing data at various resolutions, including single-cell data, multi-region data, and large-scale NGS sequencing datasets encompassing most cancer types. These analyses revealed associations that were correlated with significant differences in prognosis, demonstrating the translational implications of our approach. Furthermore, they demonstrated the versatility of our framework, which can effectively handle experimental data generated by different NGS technologies.

We finally note that, in principle, ASCETIC might accommodate any generic omic events (e.g., copy number alterations or chromatin modifications such as epigenetic changes), as long as they are heritable during cancer evolution. This capability could enable us to move beyond a genomic-centered characterization of cancer evolution. Furthermore, ASCETIC has the capability to consider not only survival data but also distinct phenotypic and biological features as potential targets for association through regularized regression. By incorporating these additional factors, ASCETIC has the potential to offer further insights into the mechanisms associated with the evolutionary steps. Exploring these possibilities is an area of future research that remains to be investigated.

## Methods
### Data model
We consider two inputs for our inference task, namely a *cross-sectional* dataset D and a temporal graph $G_P$. The cross-sectional dataset D comprises $n$ somatic alterations $V = \{1,...,n\}$ for $m$ distinct patients and is represented as an $m \times n$ binary data matrix where the rows are the patients and the columns are the somatic alterations modeled as Bernoulli random variables. For each alteration we report 1 if it is observed in each patient and 0 otherwise. Furthermore, we assume the $n$ alterations to have been selected as possible *candidate drivers*, e.g., as previously discussed[7], and, thus, $n \ll m$.

We then augment our inputs with an estimation of temporal orderings among the selected alterations as the ones that might be

obtained by building mutational tree models from single-cell or multiple biopsies data (see Supplementary Materials for details). As such, for each sample $\phi$ we have a Directed Acyclic Graph (DAG) $G_\phi = (V_\phi, A_\phi)$, where the nodes of the graph $V_\phi \subseteq V$ are the genomic alterations observed in the patient $\phi$-th and the directed arcs $A_\phi$ denotes a temporal order indicating that the parent node likely occurred earlier and the child node later during the evolution of the tumor as intended, e.g., by a mutational tree.

If we consider $\phi = 1, \ldots, m$ samples, we can extend this representation in order to consider the evolution across multiple patients. To do so, we define $G_P = (V, A)$ as the union graph of all the $G_\phi$, that is the graph containing all the alterations $V$ connected by the arcs present in at least one of the $G_\phi$, i.e., $A = \bigcup_{\phi=1}^{n} A_\phi$. In addition, this graph is augmented with a dummy node R (i.e., Root) connected toward all the alterations, which represents the wild type genotype. We notice that this construction of $G_P$ may lead to graphs containing cycles. Such property implies that these graphs do not directly describe one unique temporal ordering among events. We refer to the Supplementary Materials for a discussion on how to build such inputs from the currently available genomics data.

## Estimating the time order among driver alterations

As mentioned, the temporal graph $G_P$ may contain cycles, indicating inconsistencies in the temporal orderings inferred for each patient, i.e., it does not define a global time ordering between all events. This may be due to multiple reasons, such as real irregularities in the progressions (e.g., a gene is observed as an early mutation in one patient, but it is also observed late in another one) arising among the different patients or noisy observations which led to incorrect inference of the time orderings. Regardless of the reasons, cycles are expected to occur in practice, while analyzing the kind of data we have at hand, hence there is the need for a measure to quantify such irregularities and for an algorithm to estimate a set of time orderings which are most supported by the data (i.e., minimizing the observed inconsistencies) across multiple patients.

To this end, we consider a measure of hierarchy within a directed graph defined as follow. Given a directed graph $G = (V, A)$ and a ranking metric (e.g., in our case the time ordering of accumulation of driver alterations during tumor evolution), any arc from nodes that are "higher" in the hierarchy (e.g., alterations that occur in later stages of the tumor) to nodes that are "lower" in the hierarchy (e.g., alterations that occur at the initiation of the tumor) are not expected and they are said to be causing "agony"[8].

Although the number of possible rankings of a directed graph is exponential, Tatti and colleagues[8] provide a polynomial-time algorithm for finding a good ranking at minimum agony with respect to the union graph given as input. We here adopt this approach to compute a preferential ranking $r(.)$, i.e., a set of time orderings well supported by our observations $G_P$. Furthermore, by removing any arc inconsistent with such a ranking, we can learn from the data a set of temporal ordering that are best supported among each patient and provide an ordering among the considered driver mutations.

## Modeling selective advantage in tumor evolution

We model selective relations by enforcing a set of probabilistic constraints in our model derived by the theory of probabilistic causality[7]. The resulting network, named Suppes-Bayes Causal Network (SBCN)[10], biologically resembles the notion of accumulation of somatic alterations during cancer progression.

Specifically, in its original formulation, two probabilistic conditions are enforced in a SBCN, namely *Temporal Priority* (TP) and *Probability Raising* (PR). TP refers to the presence in the data of a temporal pattern. Let us consider a pair of event $u$ and $v$ (e.g., any pair of genetic alterations) and let us assume that we have a relation of selective advantage between them where $u$ precedes and selects for $v$.

In this case, TP implies the presence in the data of observations where $u$ often occurs before $v$, yet with possible irregularities where this is not the case. This notion can be naturally re-framed in terms of the existence of a temporal hierarchy among the two events as described before. Therefore, we can consistently evaluate when the TP condition is verified for any event toward its candidate successor gene by computing a ranking $r(.)$ as defined above and then verifying if $r(u) < r(v)$.

The PR condition subsumes instead the presence of a statistically significant pattern of occurrence between pair of observables. In particular, this adds a further meaning to the ranking between pair of events defined above: $u$ is a valid *predecessor* of $v$ if it occurs before and if a significant pattern is observed between the two events with the earlier occurrence of $u$ raising the expectation of subsequently observing $v$ as well. This condition has been proved to be equivalent to the presence of positive correlation between the events but can be efficiently verified only if we assume the presence of only conjunctive predecessors of common later events[10].

To overcome this hurdle, we here extend probabilistic causation in order to directly exploit the temporal information provided in the temporal graph $G_P$. To do so, instead of estimating the conditions on the *whole dataset* for any pair of event $u$ and $v$, we limit our analysis to consider only the subset of the data where both $u$ and $v$ are observed together. This leads us to the formulation of an extended version of the theory of probabilistic causality where we can verify the PR condition as:

$$P(u,v \mid t_u < t_v) > P(u,v \mid t_u \geq t_v) \qquad (1)$$

This is equivalent to:

$$P(t_u < t_v \mid u,v) > P(u,\bar{v}) \qquad (2)$$

This intuitively aims to understand whether, when $u$ and $v$ both happened, $u$ consistently occurred before. We refer to the Supplementary Materials for the mathematical proof.

As extensively discussed in the Supplementary Materials, this formulation allows us to define an efficient algorithm framework (ASCETIC) to perform the inference of SBCNs. The algorithm first creates a partially ordered set (poset) among the $n$ genomic alterations. This poset accounts for TP being computed by a ranking at minimum agony derived from the temporal graph $G_P$. Furthermore, it also accounts for PR as defined above. It considers each pair of event $u$ and $v$ and estimates both $P(t_u < t_v \mid u,v)$ and $P(u,\bar{v})$ from D and $G_P$. Once the poset is created, the final model can be estimated by maximum likelihood. This step aims at reducing the presence of false positives, yet possibly leading to some false negative.

## Tests on simulated data

We extensively assessed the performance of our method on simulated data with different configurations, with particular focus of comparing the evolutionary models output of ASCETIC with the ones inferred by competing methods. We considered input data for multiple patients coming both from NGS sequencing of a single biopsy per tumor and single-cell sequencing or bulk sequencing NGS data providing multiple samples for the same tumor.

For the first scenario, we randomly generated 100 weakly connected directed acyclic graphs of density 0.4 and 10 nodes (Bernoulli variables with value 0 or 1). We generated samples from the distribution induced by such *DAGs* constraining for a cumulative model[10,65]. We repeated these experiments for the case of networks of conjunctive *AND* parent sets, disjunctive *OR* parent sets and exclusive disjunctive *XOR* parent sets. Furthermore, we considered datasets of sample size 50, 100, and 200 patients and we added noise to the data including random entries at different probabilities to model any error in the generation of the data. We considered noise of levels 0%, 5%,

10%, 15%, and 20%. This led us to a total of 4500 randomly generated cross-sectional datasets.

For each dataset, we then simulated *cancer cell fractions* (CCF). To do so, we randomly generated a total order consistent with the generative structure for each (possibly noisy) sample. This total order may represent a set of time observations such as follow-ups through time of patients. For each node in the ordering, we generated a random value within (0,1) with higher values representing cancer cell fractions of early mutated genes and lower values representing late mutations. We then added noise to the simulated CCF by sampling from a Gaussian distribution with mean being the simulated CCF and variance of 0.00, 0.01, 0.05, 0.10, and 0.20. With this, we obtained 22,500 configurations for the NGS sequencing of a single biopsy data type.

We also considered a simple model of single-cell sequencing and bulk sequencing NGS data providing multiple samples for the same tumor. In this case, we simulated the progression for an individual patient as a tree with different generative models to generate single-cell or bulk sequencing data[65]. Also in this case we repeated the experiments for 100 independent runs and for each of them we generated a progression model as a composition of 3 individual-level progressions with independent or overlapping events. In the first case, our random structures consisted of 3 trees of 5 genes each (structures of 15 nodes) without any overlap among the genes (i.e., a gene in tree 1 is not present in neither tree 2 nor 3). In the second case, we still generated 3 trees of 5 genes each but this time we randomly overlapped 5 of the genes, leading to topologies of 10 nodes. In the case of bulk sequencing, we generated 10 biopsies for either 10 or 20 patients (i.e., cross-sectional datasets of 100 or 200 entries) and adopted noise levels of 0%, 1%, 5%, 10%, and 20%. In the case of single-cell sequencing, we generated 25 cells for either 5 or 10 patients (i.e., cross-sectional datasets of 125 or 250 entries). In this case, as previously done[65], we used unbalanced noise levels for false positive and false negative (allele dropout) errors respectively of 0%, 1%, 2%, 3% and 4%, and 0%, 10%, 20%, 30%, and 40%. This led to a total of 4000 randomly generated cross-sectional datasets. In this case we did not generate CCFs, but we inferred the evolutionary model from the simulated data of each patient with the variant of Edmonds' algorithm proposed in the TRaIT framework[65].

We evaluated the results for each configuration comparing ASCETIC, the CAPRI algorithm[10] and the standard maximum likelihood fit approach for structure learning in terms of true positives (TP), true negatives (TN), false positives (FP) and false negatives (FN) with respect to the generative ground truth model. Specifically, we defined TP the arcs both in the generative model and in the inferred ones. Similarly, TN were the arcs missing in both the models. FP and FN were respectively the arcs inferred by the algorithms but not in the generative model and the arcs not inferred which were in the generative model. Given these measures, we assessed the results in terms of accuracy ($\frac{TP+TN}{TP+TN+FP+FN}$), precision ($\frac{TP}{TP+FP}$), recall (or sensitivity) ($\frac{TP}{TP+FN}$) and specificity ($\frac{TN}{TN+FP}$).

## Processing single-cell sequencing data
We considered single-cell data generated with the Tapestri Sequencing Platform for a set of myeloid malignancies[15]. The dataset provides single-cell mutational profiles of 146 samples from 123 distinct patients. Following the analysis provided in the original paper[15], as a quality filter, we considered only samples with at least 100 single cells and 2 mutations. We built D for all the samples by reporting 1 if a certain mutation was observed in at least one cell for each patient. $G_P$ was instead obtained by running ∞SCITE[66] on the single-cell data independently for each sample with the following parameters for MCMC execution and noise estimation: -r 10 -l 1000000 -fd 0.01 -ad 0.10 -cc 0.001 -s -p 10000 -e 0.2.

## Processing multi-region sequencing data
We considered the data from the TRACERx research project[36] comprising multi-region sequencing data for a total of 302 biopsies from 100 distinct patients affected by lung cancer and a total of 65421 somatic substitutions. We built D for all the samples by reporting 1 if a certain mutation was observed in at least one of the multi-regional samples for the patient. $G_P$ was instead obtained directly from the models inferred by the REVOLVER method[6]. We selected for the ASCETIC analysis genes occurring in at least 3 samples and samples with at least 2 variants.

## Processing single biopsy NGS sequencing data
We considered the data from the Pan-Cancer Atlas[32] and the MSK-MET[41] studies for a total of more than 35,000 distinct patients across most cancer types. Specifically, we selected for the analysis all the patients where both somatic mutations and copy number alterations data were available and cancer types with at least 50 samples. We then considered for each cancer type the top mutated genes from a list of known cancer-related genes[67-69] that were observed at least in 1% of the samples, for a maximum of 15 candidate driver mutations per cancer type (our input matrix D). $G_P$ was estimated considering cancer cell fractions CCFs data computed from read counts adjusted for copy number data. Specifically, to derive CCFs we utilized variant allele frequencies data obtained from coverage information. These frequencies were normalized with an estimate of the number of copies in the patients' genomes based on copy number data, relative to the expected normal ploidy. The copy number data utilized for the analysis were obtained from Affymetrix SNP6 for the Pan-Cancer Atlas studies and from targeted sequencing via MK-IMPACT for the MSK-MET dataset. We report more details in the Supplementary Materials.

## ASCETIC parameters for the analyses of real data
In order to improve the performance and the stability of ASCETIC on real data, we adopted a re-sampling procedure for a robust estimation of the *r(i)* rankings. For the analysis of multi-region sequencing data, we performed 100 bootstrap iterations where we sampled our inputs D and $G_P$ and estimated rankings among genes each time; we then used for the subsequent steps of ASCETIC, the mean (rounded to integer) of these 100 rankings. For the analysis of single biopsy NGS sequencing data, we adopted a similar idea, but this time, we performed re-sampling for a robust estimation of cancer cell fractions by sampling from a beta distribution given the read counts for each sample; also in this case, we iterated this procedure 100 times and we used the mean (rounded to integer) of the 100 rankings as input to ASCETIC. The likelihood fit step of ASCETIC was performed with the hill climbing algorithm with 100 restarts using AIC as a model selection criterion. We finally performed cross-validation to estimate the confidence of each temporal relation returned by our method. This was done using an 80−20 split and 100 repetitions of the ASCETIC pipeline.

## Survival analysis
To assess significant differences in prognosis implied by the evolution models returned by ASCETIC, we implemented a combined regularized Cox regression and Kaplan−Meier survival analysis[70]. In particular, we considered all the parental relations between driver mutations inferred by ASCETIC as input covariates for this analysis.

For each tumor, we selected the set of alterations associated with the minimum cross-validation error and stratified the patients into different risk groups. The survival curves of the different risk groups were finally assessed via standard Kaplan−Meier analysis.

## Reporting summary
Further information on research design is available in the Nature Portfolio Reporting Summary linked to this article.

## Data availability

All cancer data used in this study are publicly available from the relative original publication or from the cBioPortal repository (https://www.cbioportal.org/). In particular, the Acute Myeloid Leukemia dataset from ref. 24 can be downloaded at https://www.cbioportal.org/study/summary?id=aml OHSU 2018. Moreover, the Pan-Cancer Atlas datasets and the MSK-MET datasets can be downloaded from the cBioPortal repository at https://www.cbioportal.org/datasets. The list of the Pan-Cancer Atlas datasets is the following:

Acute Myeloid Leukemia (AML). Adrenocortical Carcinoma (ACC). Bladder Urothelial Carcinoma (BLCA). Brain Lower Grade Glioma (LGG). Breast Invasive Carcinoma (BRCA). Cervical Squamous Cell Carcinoma (CESC). Cholangiocarcinoma (CHOL). Colorectal Adenocarcinoma (COADREAD). Diffuse Large B-Cell Lymphoma (DLBC). Esophageal Adenocarcinoma (ESCA). Glioblastoma Multiforme (GBM). Head and Neck Squamous Cell Carcinoma (HNSC). Kidney Chromophobe (KICH). Kidney Renal Clear Cell Carcinoma (KIRC). Kidney Renal Papillary Cell Carcinoma (KIRP). Liver Hepatocellular Carcinoma (LIHC). Lung Adenocarcinoma (LUAD). Lung Squamous Cell Carcinoma (LUSC). Mesothelioma (MESO). Ovarian Serous Cystadenocarcinoma (OV). Pancreatic Adenocarcinoma (PAAD). Pheochromocytoma and Paraganglioma (PCPG). Prostate Adenocarcinoma (PRAD). Sarcoma (SARC). Skin Cutaneous Melanoma (SKCM). Stomach Adenocarcinoma (STAD). Testicular Germ Cell Tumors (TGCT). Thymoma (THYM). Thyroid Carcinoma (THCA). Uterine Carcinosarcoma (UCS). Uterine Corpus Endometrial Carcinoma (UCEC). Uveal Melanoma (UVM).

The MSK-MET dataset can be downloaded at MSK-MET. The remaining data are available within the Article, Supplementary Information or Source Data file. Source data are provided with this paper.

## Code availability

ASCETIC is available as an R package on GitHub (https://github.com/danro9685/ASCETIC).

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

## Acknowledgements

This work was partially supported by a Bicocca 2020 Starting Grant to D.R. and F.A. and by the Italian Ministry of University and Research (MIUR)—Department of Excellence project PREMIA (PREcision MedIcine Approach: bringing biomarker research to the clinic) to R.P. Support was also provided by the Cancer Research UK and Associazione Italiana per la Ricerca sul Cancro (CRUK/AIRC) "Accelerator Award" (award number 22790) "Single-cell Cancer Evolution in the Clinic" to A.G., G.C., and M.A. G.C. acknowledges funding from the Italian Foundation for Cancer Research (AIRC) under MFAG 2020, ID 24913 project. L.M. acknowl-edges funding from the Italian Foundation for Cancer Research (AIRC) under IG 2020, ID 24828 project. Partial support to A.G. and M.A. was also provided by the MUR under the grant "Dipartimenti di Eccellenza

2023–2027" of the Department of Informatics, Systems and Communication of the University of Milano-Bicocca, Italy. M.A. acknowledges funding from "European Commission Program PPPA2027, PPPA-2021-AIPC #LC-01815952/101052609, 'Towards an UNIque approach for artificial intelligence data-driven solutions to fight Childhood cAncer FOR Europe, UNICA4E".

## Author contributions

Conceptualization: D.R. and A.G. Methodology: D.R. Software: D.R. and L.D. Investigation: D.F., I.C., V.C., F.M., M.V., A.A., L.M., and D.R. Visualization: F.A., A.G., and D.R. Funding acquisition: F.A., M.A., R.P., G.C., A.G., L.M., and D.R. Supervision: M.A., G.C., R.P., A.G., L.M., and D.R. Writing—original draft: D.F., I.C., V.C., F.M., M.V., A.G., L.M., and D.R. All authors read and approved the final manuscript.

## Competing interests

The authors declare no competing interests.
