## [Peer Review File · Nature Communications]

Evolutionary signatures of human cancers revealed via genomic analysis of over 35,000 patientsReviewers' Comments:

Reviewer #1:

Remarks to the Author:

The authors present a methodology to infer order of acquisition of driver mutations given a dataset of patients with single-cell or bulk sequencing data available from one or more tumors. They then apply their simulated data to demonstrate its performance in comparison to two other methods: CAPRI and a standard maximum-likelihood procedure. Finally, they analyze multiple cancer datasets. The manuscript is of potential interest to the cancer genomics field

In the analysis for AML, four evolutionary signatures are extracted. In the first two steps, the algorithm identifies the evolutionary trajectories underlying the co-mutation matrix defined by the most significant orderings of drivers. In the next step, the algorithm uses the survival data to select the most relevant gene-gene interactions to cluster to patients into risk groups. In the final step, survival probability of different patient groups is compared. My question in relation to this method are as follows:

1. How much of an advancement does the new method provided in comparison to the authors old method, CAPRI, given that their performance are similar.
2. For the AML analysis, the identified evolutionary signatures seems to overlap in the most prevalent features. For example, NPM1 mutation without any other driver are present in 50% of the cases that fit signatures 1 and 3. This is very counter-intuitive given that sig-1 and -3 are on the lower end and higher end of the risk spectrum. Shouldn't the features in each cluster be distinct if these were real clusters?
3. The most relevant driver orderings in evolutionary signatures is selected using survival data. Isn't this a bit circular?
4. It was not clear to us if CCFs is taken into account? If yes, how is it calculated? What kind of copy number data is utilized?
5. Could this method be used to extract evolutionary patterns shared across many cancer types?

Reviewer #2:

Remarks to the Author:

Fontana and colleagues present ASCETIC, a framework that extracts evolutionary trajectories (or signatures) from NGS experiments. They apply their tool to a large number of different datasets and demonstrate that their evolutionary subgroups hold prognostic relevance.

The concept of harnessing NGS data to reveal novel biology and potentially important patient subgroups for patient stratification is an important task. I think a key challenge of the paper is 1) whether ASCETIC really defines evolutionary trajectories; 2) what is the real purpose of these signatures? 3) If vastly different results are obtained depending on the data type, is it really meaningful?

Because of the issue above, I have not focussed my review on the details of the ASCETIC model, rather I have focussed on higher level issues, which I think need resolving first, before the method itself is truly evaluated.

Specific comments:

I am unclear whether the key goal of ASCETIC is to identify the true evolutionary histories of tumours, group these, and then identify common evolutionary signatures or trajectories of tumours, or whether

it is to identify genomically encoded groups of tumours with clinical relevance. The authors present the tool as achieving both of these, but I think they are quite distinct. If the true purpose is to identify the underlying biology, the authors should focus on this and not necessarily survival. Conversely, if the focus is survival, much more validation is required. If you look hard enough you can always find an association with survival. But is it spurious or does it really reflect something prognostic? The current manuscript does not address this.

Given the important caveats above, I think if the authors wish to truly present an analysis that attempts to link evolutionary subgroups with survival, much more validation is needed:

For each subtype a validation cohort is needed

The premise of the manuscript is that the survival grouping is improved by using the 'evolutionary signatures', however, there is a need to include further validation of this assumption. For instance if simply mutation data is included (without an order), can similar groups be deduced? The authors state information to this effect is in Supplementary Figures 17-18, but I couldn't find this?

The authors seek to validate ASCETIC's evolutionary steps on acute myeloid leukemia samples using unseen single-cell data from 123 AML patients. The authors state that "Notably, most of the evolutionary steps returned by ASCETIC were highly consistent with the mutational trees." However no statistical data is presented in the main text. And, the table in the supplementary is rather confusing.

The authors apply their tool to different data-sets. However, each data-set is treated in a very different way. If this is to be a useful tool, the authors should explore how robust it is. E.g. how different do the results look for TRACERx LUAD vs. MSK LUAD or TCGA LUAD?

To expand on the point above, how much does it impact the outcome of the model using single cell vs single-region bulk vs multi-region bulk? Presumably the inferred phylogenies could be very different based on the data type, which will massively impact the input data to ASCETIC, and consequently the downstream predicted survival risk group? To follow this, how much would sampling bias affect the results of ASCETIC?

The ASCETIC model heavily relies on the previously annotated driver mutations for the input data. This means that ASCETIC assumes that these annotated driver mutations within a tumour are indeed having a functional effect on the growth of the tumour, and, conversely assumes that the subclones with no annotated driver mutation are solely arising due to drift and do not have an impact on the clonal dynamics of tumour growth. This may be a significant limitation of the model, as it ignores variation in clonal dynamics or indeed other genomic variance such as copy number alterations, which may both contribute to survival outcome.

The authors present a validation of ASCETIC to show the accuracy of their method on simulated data, however this validation does not support an association of the detected evolutionary signatures with survival outcome. It is also not entirely clear whether the simulated data itself was created specifically for ASCETIC validation. I would suggest further validation of the evolutionary signatures identified. The font is also incredibly small in the supplementary figures.

Reviewer #1

The authors present a methodology to infer order of acquisition of driver mutations given a dataset of patients with single-cell or bulk sequencing data available from one or more tumors. They then apply their simulated data to demonstrate its performance in comparison to two other methods: CAPRI and a standard maximum-likelihood procedure. Finally, they analyze multiple cancer datasets. The manuscript is of potential interest to the cancer genomics field.

We would like to thank the reviewer for their positive feedback on our work and its significance to the cancer research community. We also appreciate their valuable comments regarding the need to better discuss the differences and novel aspects of the ASCETIC framework compared to other methods, such as CAPRI, for analyzing cancer evolution from genomic data. While both ASCETIC and CAPRI share some similarities in inferring evolutionary models of cancer, ASCETIC distinguishes itself as the only algorithmic framework that can directly associate these evolutionary models with prognosis. In the revised version of the manuscript, we now provide a clear description of this unique feature of ASCETIC.

Next, we present a thorough and detailed response to each of the points raised by the reviewer.

In the analysis for AML, four evolutionary signatures are extracted. In the first two steps, the algorithm identifies the evolutionary trajectories underlying the co-mutation matrix defined by the most significant orderings of drivers. In the next step, the algorithm uses the survival data to select the most relevant gene-gene interactions to cluster to patients into risk groups. In the final step, survival probability of different patient groups is compared. My question in relation to this method are as follows:

1. How much of an advancement does the new method provided in comparison to the authors old method, CAPRI, given that their performance is similar.

We thank the reviewer for raising this issue. ASCETIC introduces several significant theoretical and practical advancements in comparison to CAPRI and existing methods. We acknowledge that these advancements may not have been sufficiently explained in the previous version of the manuscript. In brief, our ASCETIC framework includes two algorithmic steps and related outputs:

1. In the first step, ASCETIC performs the inference of a cancer evolution model returning the repeated evolutionary trajectories that are consistently observed across patients. Despite the goal of this step of ASCETIC is akin to that of CAPRI, three major differences can be highlighted:
 - a. The expressivity of ASCETIC is significantly superior, as CAPRI (and most existing methods) allows one to infer only conjunctive relations among genomic events (e.g. $[A \text{ and } B] \rightarrow C$), whereas our new method relaxes this hypothesis as it returns partial orderings among genes and can model any type of parental relation. This aspect is a key feature of our approach, enabling ASCETIC to provide precise and dependable models of cancer evolution.
 - b. The applicability of ASCETIC is superior to that of CAPRI, as the ASCETIC framework is designed to accommodate sequencing data generated from: (bulk) single biopsies, (bulk) multi-region biopsies, or even single-cell data, whereas CAPRI can take as input only bulk sequencing data from single biopsies. This aspect dramatically enhances the usability of our new method in real-world datasets with different modalities.

- c. ASCETIC consistently outperforms CAPRI across various metrics in simulated scenarios. Specifically, ASCETIC emerges as the top-performing method on average across all settings and the 8,500 generated synthetic cancer evolution models.
2. In the second step, ASCETIC associates the inferred trajectories with survival data to predict those evolutionary steps that could have potential clinical significance. Such evolutionary steps are named as single-nucleotide variants (SNV) evolutionary signatures and can be exploited to stratify patients.
This step is *NOT* present in CAPRI (or in any competing method) and presents one of the major novelties of the work, with a high potential impact in future clinical practices.

We agree with the reviewer that, in the previous version of the manuscript, we did not adequately highlight the significant new features introduced by ASCETIC as compared to state-of-the-art tools for analyzing cancer evolution from genomic data. We have addressed this issue in the revised version of the manuscript by providing a clear and detailed discussion of these enhanced features.

2. For the AML analysis, the identified evolutionary signatures seem to overlap in the most prevalent features. For example, NPM1 mutation without any other driver are present in 50% of the cases that fit signatures 1 and 3. This is very counter-intuitive given that sig-1 and -3 are on the lower end and higher end of the risk spectrum. Shouldn't the features in each cluster be distinct if these were real clusters?

In the AML study, both signatures 1 and 3 are characterized by mutations in the NPM1 gene, which are significantly associated to clinical outcomes. However, in signature 3, mutations in the FLT3 gene are also observed and significantly associated with survival. In particular, ASCETIC infers that the evolutionary step from NPM1 to FLT3 is also associated with (poor) prognosis. These two signatures resemble known subtypes in AML, where patients with only NPM1 mutations are shown to have a better prognosis compared to patients with both NPM1 and FLT3 co-mutations.

In the paragraph "Stratification Based Solely on Mutations" in the main text, we describe the same analysis performed by using only single mutations as features, and we show that these two subtypes cannot be distinguished. This proves that the evolutionary model returned by ASCETIC indeed provides additional information that can be exploited to stratify patients in distinct risk groups.

In summary, ASCETIC returns evolutionary steps that might be significantly associated with clinical outcomes and that can be present in multiple cancer subtypes. This is modeling repeated cancer evolutions showing common elements, but still different progressions and might lead to a more comprehensive understanding of cancer evolution. We have now clarified this aspect in the paper.

3. The most relevant driver orderings in evolutionary signatures is selected using survival data. Isn't this a bit circular?

As mentioned in a previous answer to the reviewer, ASCETIC generates two main outputs: (1) a model that captures the evolutionary trajectories that are consistently repeated during tumor evolution in different patients, and (2) a set of genomic features, defined as SNV evolutionary signatures, that are significantly associated to clinical outcomes, and which can be exploited to stratify patients.

The second step is performed subsequently to the first one, considering the evolutionary steps as features of the regularized Cox regression model. In other words, ASCETIC does not directly derive

the evolutionary signatures from survival data, rather, it first infers such orderings from genomic data and subsequently associates them with outcome. So, there are no circularity issues. We have now provided a better clarification on this aspect in the manuscript.

4. It was not clear to us if CCFs is taken into account? If yes, how is it calculated? What kind of copy number data is utilized?

When using cross-sectional bulk datasets, our framework exploits cancer cell fractions (CCFs) to establish a partial ordering of the driver genes in each patient, orderings which are then used as input for the subsequent steps of the analysis. In particular, to derive CCFs, we use variant allele frequencies, normalized using an estimate of the number of copies in the patients' genomes. The copy number data used for normalization are obtained from Affymetrix SNP6 for the PanCancerAtlas studies and from targeted sequencing via MK-IMPACT for the MSK-MET dataset.

This is described in Section 1.2 of the Supplementary Materials, titled "The Evolution of Cancer as a Graph".

To improve the overall comprehensibility, we have revised the "Processing Single Biopsy NGS Sequencing Data" paragraph and the main text's Methods Section by providing a description of how CCFs are computed. We would like to thank the reviewer for bringing to our attention the need to improve this part, which in the previous version of the manuscript was not clearly specified.

5. Could this method be used to extract evolutionary patterns shared across many cancer types?

A common practice in statistical inference is to consider homogenous groups separately, in order to reduce the impact of sample heterogeneity on inference accuracy and robustness (see, e.g., the well-known Simpson's paradox, phenomenon that can occur when a trend or relationship observed within different groups of data reverses or disappears when the same groups are combined). This is the reason why, typically, cancer evolution models are derived from stratified cohorts (e.g., molecular subtypes or gene expression clusters).

Accordingly, it is sound to apply ASCETIC to samples from different cancer subtypes and return single evolutionary models.

However, we completely agree with the reviewer that highlighting/discovering possible regularities or evolutionary trajectories shared across tumor types might be useful, especially from the translational perspective.

For this reason, we have now included in the revised manuscript (Supplementary Table 6) a consensus evolutionary model of all the 24 cancer subtypes from the PanCancerAtlas studies and all the 41 cancer subtypes from the MSK-MET dataset, in which the evolutionary trajectories are weighted upon the occurrence across cancer types.

We believe that this is an additional important result of our work, for which we thank again the reviewer.

Reviewer #2

Fontana and colleagues present ASCETIC, a framework that extracts evolutionary trajectories (or signatures) from NGS experiments. They apply their tool to a large number of different datasets and demonstrate that their evolutionary subgroups hold prognostic relevance.

The concept of harnessing NGS data to reveal novel biology and potentially important patient subgroups for patient stratification is an important task. I think a key challenge of the paper is 1) whether ASCETIC really defines evolutionary trajectories; 2) what is the real purpose of these signatures? 3) If vastly different results are obtained depending on the data type, is it really meaningful? Because of the issue above, I have not focused my review on the details of the ASCETIC model, rather I have focused on higher level issues, which I think need resolving first, before the method itself is truly evaluated.

We would like to thank the reviewer for recognizing the importance of ASCETIC's goals for the cancer research community. Moreover, we appreciate the valuable feedback regarding the need for a clearer presentation of our method's objectives, particularly with regards to the semantics of the proposed evolutionary signatures.

Furthermore, we acknowledge that demonstrating the reproducibility of our results is important, as it is for all computational methods commonly used in the analysis of genomic data. While the availability of datasets for certain cancers may be limited, we have taken every opportunity to include new validations on external data to support the validity and generalizability of our approach, which we will detail in the following.

We provide a detailed point-by-point response to each issue raised by the reviewer and outline the new analyses that have been included in the revised manuscript.

Specific comments:

I am unclear whether the key goal of ASCETIC is to identify the true evolutionary histories of tumours, group these, and then identify common evolutionary signatures or trajectories of tumours, or whether it is to identify genomically encoded groups of tumours with clinical relevance. The authors present the tool as achieving both of these, but I think they are quite distinct. If the true purpose is to identify the underlying biology, the authors should focus on this and not necessarily survival. Conversely, if the focus is survival, much more validation is required. If you look hard enough you can always find an association with survival. But is it spurious or does it really reflect something prognostic? The current manuscript does not address this.

We thank the reviewer for raising this issue, which fostered a fruitful discussion among the authors regarding the final goal of our framework and its presentation.

We agree with the reviewer that the inference of cancer (genomic) evolution and the identification of “prognostic biomarkers” are separate topics, especially because the relation among them is still somehow undeciphered.

Indeed, this is the main motivation underlying the development of our framework, which translates into a simple question: *“is it possible to verify whether certain repeated patterns of genomic evolution observed across cancer patients are consistently associated to better or worse prognoses? Thus, improving over predictions made leveraging single genetic alterations only?”*

To answer to this question, our framework:

1. First, reconstructs robust models of cancer evolution, but without employing survival data.
2. Second, it employs the identified repeated evolutionary trajectories as features for a regularized Cox regression model. Only those trajectories – i.e., sets of genomic alterations - that are significantly associated with the survival outcome are returned.

In other words, ASCETIC allows one to perform a model-informed feature selection, which also highly enhances the interpretability of the results, providing an indication on where the prognostic indicators are positioned along the temporal evolution of the disease.

Surprisingly, many of such significant patterns – named as SNV evolutionary signatures – are found in most tumor types, which we believe it's an important theoretical and practical novelty of our approach.

To the best of our knowledge, no other method allows one to perform a similar analysis.

As suggested by the reviewers, it could be valuable to consider additional phenotypic/biological factors, beyond survival, as target variables for regression analyses. However, given the scarcity of clinical covariates available in large genomic datasets, exploring these factors falls outside the scope of our work. Consequently, a comment addressing this point has been included in the manuscript.

We concur with the reviewer's observation that our previous manuscript version did not adequately elucidate the significant new features and goals that ASCETIC introduces when compared to the current state-of-the-art tools for analyzing cancer evolution from genomic data. Therefore, we have now revised our manuscript to provide a more comprehensive and detailed description and discussion of ASCETIC's goals and outputs.

Moreover, we included several new experiments to further validate the results delivered by our approach (see the revised main text and supplementary figures 89 to 108 at pages 107 to 126), including:

- 1) Validation of ASCETIC Evolutionary Signatures and Subgroups for AML on an external dataset from the TCGA studies.
- 2) Validation of ASCETIC Evolutionary Signatures and Subgroups for Metastatic Lung Adenocarcinoma (MSK-MET) on an external dataset from the TCGA studies.
- 3) Validation of ASCETIC Evolutionary Signatures and Subgroups for Metastatic Lung Squamous Cell Carcinoma (MSK-MET) on an external dataset from the TCGA studies.
- 4) Validation of ASCETIC Evolutionary Signatures and Subgroups for Prostate Cancer (MSK-MET) on two external datasets, one from the TCGA studies (only primary cancers) and one from the Stand Up To Cancer (SU2C) initiative (only metastatic cancers).
- 5) We performed an additional new analysis where we performed the ASCETIC framework on three distinct datasets for Gliomas. The framework returned very consistent Evolutionary Signatures and Subgroups for all the three datasets.

All these new results have been included in the manuscript's main text and supplementary materials, providing compelling evidence to substantiate the robustness and significance of ASCETIC.

We believe that with these improvements, our manuscript now provides a clear description of the objectives and significance of our approach.

Given the important caveats above, I think if the authors wish to truly present an analysis that attempts to link evolutionary subgroups with survival, much more validation is needed:

For each subtype a validation cohort is needed.

The premise of the manuscript is that the survival grouping is improved by using the 'evolutionary signatures', however, there is a need to include further validation of this assumption. For instance if simply mutation data is included (without an order), can similar groups be deduced? The authors state information to this effect is in Supplementary Figures 17-18, but I couldn't find this?

Following the suggestion by the reviewer, we included a series of new experiments and validations, that are detailed in our previous reply and discussed in the manuscript.

Regarding the second point raised by the reviewer, we apologize if the specific analyses comparing the stratification based on single genomic alterations versus the evolutionary steps identified by ASCETIC were not clearly conveyed in the manuscript. These analyses are included in the main text, specifically in the paragraph titled "Stratification based solely on mutations." Supplementary Figures 17-18 on pages 30 and 31 of the Supplementary materials also present the subtypes and their associated survival curves.

It is important to highlight that the analysis using only mutation data failed to identify certain crucial molecular features of AML, such as the NPM1 to FLT3 co-mutation, which is a known prognostic marker. Furthermore, due to the lack of information regarding the parent event for NRAS, it was not possible to associate this gene with specific prognostic groups. In contrast, ASCETIC successfully associated different evolutionary trajectories with NRAS as a late event in AML evolution, which, depending on the preceding mutations (DNMT3A, ASXL1, or JAK2 genes), resulted in distinct evolutionary signatures with significantly different prognoses (evolutionary signature #2 and #4). Therefore, our findings highlight the limitations of relying solely on mutation data and emphasize the importance of considering the evolutionary history of tumors for a more informative classification of cancer subtypes.

We have revised the manuscript to improve the description of these results.

The authors seek to validate ASCETIC's evolutionary steps on acute myeloid leukemia samples using unseen single-cell data from 123 AML patients. The authors state that "Notably, most of the evolutionary steps returned by ASCETIC were highly consistent with the mutational trees." However no statistical data is presented in the main text. And, the table in the supplementary is rather confusing.

We apologize for the lack of clarity of the Supplementary Table on page 105 of the Supplementary Material, which may have caused confusion. The purpose of the additional analysis presented in the table is to validate the evolutionary model inferred by ASCETIC, which shows the evolutionary steps that are consistent across multiple AML patients.

To this end, we employed an additional single-cell dataset by Morita et al., in which the authors performed a curated phylogenetic analysis to recapitulate the evolutionary history of each patient individually, yet without providing a model of evolution common to the patients.

In detail, we used the phylogenies from that work to validate the 17 evolutionary steps identified by ASCETIC. In the Supplementary Table, we report for each of these 17 arcs: (1) the number of phylogenies/patients where the same arc is reported (column $P \rightarrow C$), (2) the number of phylogenies/patients where an arc inconsistent with the one of ASCETIC is reported (column $C \rightarrow P$, where the predecessor gene P is inferred in the phylogeny to come after the gene C, in an opposite way of ASCETIC), and (3) the number of phylogenies/patients where no relation in support of the evolutionary step is provided.

Our results indicate that 15 out of 17 evolutionary steps by ASCETIC are consistent with the phylogenetic analysis by Morita et al., with the column $P \rightarrow C$ clearly showing the highest number of consistent phylogenies, thus proving the effectiveness of our approach.

We appreciate the reviewer for bringing to our attention that the previous version of the manuscript did not provide a clear explanation of this analysis. We have taken this feedback seriously into consideration and have now made improvements in the manuscript to address this issue.

The authors apply their tool to different data-sets. However, each data-set is treated in a very different way. If this is to be a useful tool, the authors should explore how robust it is. E.g. how different do the results look for TRACERx LUAD vs. MSK LUAD or TCGA LUAD?

Robustness and generality are key features of our framework, which are inherited by the various state-of-the-art methods employed in the distinct algorithmic steps, such as bootstrap and resampling, and which ensure the highest standards for both (i) the inference of cancer evolution model, (ii) the identification of prognostic features.

Importantly, our framework was designed to natively accommodate data generated at different resolution (single-biopsy, multi-region, single-cell), which is a complete novelty in the field.

However, we agree with the reviewer that, to provide further evidence supporting the existence of the SVN evolutionary signatures, one should assess their consistency across datasets.

Unfortunately, the three datasets specifically mentioned by the reviewer for lung adenocarcinoma represent different stages and types of lung cancer. Consequently, using them straightforwardly for the intended purpose is not feasible. In particular,

- the TRACERx dataset comprises only early-stage non-small cell lung cancers,
- MSK LUAD provides data on very advanced, metastatic lung adenocarcinomas typically subject to therapy,
- TCGA LUAD provides data on lung adenocarcinomas at diagnosis.

Due to this inherent dissimilarity, the three datasets cannot be directly compared. It is especially important to note that the TRACERx dataset includes significantly different patients compared to the others.

To address this issue, we included in the manuscript an external dataset for lung adenocarcinoma, incorporating data from three additional studies (Chen, Jianbin, et al., Vanguri, Rami S., et al., and Zhang, Tongwu, et al.).

Such dataset was employed as a validation cohort for the analysis performed for MSK LUAD. In detail, we designed a random forest classifier considering the SNV evolutionary signatures as input, and which confirmed the robustness of ASCETIC's results (Supplementary Figures 21-22 at pages 33 and 34 of the supplementary material).

In addition, based on the reviewer's comments, we also tried to further validate the evolutionary models inferred by ASCETIC on MSK LUAD by verifying them in the TCGA LUAD dataset. Despite the fact that these datasets consist of very heterogeneous patients at different stages of the disease, it is worth noting that ASCETIC has been successful in producing highly consistent results. This further underscores the robustness of our approach.

We thank the reviewer for recommending this additional validation, which we now report in the manuscript.

Finally, as specified in our previous replies, we included a set of additional validations for the other considered cancer datasets, which are outlined in a previous answer to the reviewer.

To expand on the point above, how much does it impact the outcome of the model using single cell vs single-region bulk vs multi-region bulk? Presumably the inferred phylogenies could be very different based on the data type, which will massively impact the input data to ASCETIC, and consequently the downstream predicted survival risk group? To follow this, how much would sampling bias affect the results of ASCETIC?

We completely agree with the reviewer that assessing the impact of sampling biases and resolution is an important aspect to investigate.

To this end, we would like to highlight that the ASCETIC framework has been intentionally designed to be modular. This unique feature enables the utilization of diverse state-of-the-art phylogenetic tools for inferring mutational trees, which subsequently serve as input to ASCETIC.

In addition, our method implements a bootstrap and resampling scheme to improve the statistical robustness of the results. Therefore, while all computational methods may be affected by sampling bias or noise/resolution issues, our tool utilizes advanced statistical approaches to mitigate this issue. Furthermore, ASCETIC offers a cross-validation score for each returned evolutionary trajectory. This valuable feature ensures a thorough evaluation of its reliability and provides a direct estimation of the uncertainty associated with the results.

In order to quantitatively assess the robustness of the results delivered by ASCETIC across sequencing protocols/platforms, we conducted extensive simulations, whose results are presented in Supplementary Figure 2 on page 14 of the supplementary material and subsequent Supplementary Figures 3-15.

These analyses serve as strong evidence supporting the stability and reliability of ASCETIC's results, even when variations exist in the input data.

Furthermore, the analysis of acute myeloid leukemia presented in the main text aimed to provide additional confirmation of the effectiveness of our approach in producing consistent results. In this analysis, we utilized *single-cell data* to infer the AML evolutionary model. The evolutionary steps identified were then validated using another external single-cell dataset. Subsequently, we associated these evolutionary steps with survival data obtained from a separate large-scale *bulk dataset*, which included curated survival data. The successful integration of real-world data generated using different protocols demonstrates the robustness and coherence of our approach in constructing reliable models. This finding further strengthens the validity and applicability of our methodology.

We have included a critical description of all these aspects in the main text.

The ASCETIC model heavily relies on the previously annotated driver mutations for the input data. This means that ASCETIC assumes that these annotated driver mutations within a tumour are indeed having a functional effect on the growth of the tumour, and, conversely assumes that the subclones with no annotated driver mutation are solely arising due to drift and do not have an impact on the clonal dynamics of tumour growth. This may be a significant limitation of the model, as it ignores variation in clonal dynamics or indeed other genomic variance such as copy number alterations, which may both contribute to survival outcome.

We agree with the reviewer that many other genomic and epigenetic factors are involved in the evolution of most cancer types. However, the current choice of employing only single-nucleotide variants has a many-fold motivation.

First, this allows to leverage the state-of-the-art algorithms to infer cancer evolution from (bulk and single-cell) sequencing data, which typically take SNVs as input. For example, analyzing both SNVs and larger genomic alterations such as indels or copy numbers in single-cell data remains a significant challenge. To the best of our knowledge, there are currently no statistical inference methods that effectively account for all these types of alterations together. However, given the modularity and generality of the ASCETIC framework, no theoretical impediments exist that prevent the inclusion of future methods for the inference of evolution models, including arbitrary genomic alterations. We included a comment on this issue in the main text.

Second, SNVs present a noteworthy advantage with respect to other structural variants, i.e., they can be included in targeted panels, which are common practice in most clinical settings. Given that one of the primary objectives of our work is to identify evolutionary signatures linked to prognosis, with the aim of enhancing diagnosis, it is crucial to highlight that this aspect could have significant translational implications in terms of cost-effectiveness and broader applicability. In this regard, we also recall that, despite the different objectives, the various works on mutational signatures (<https://cancer.sanger.ac.uk/signatures/>) started by analyzing SNVs, moving towards larger structural variants only successively.

Third, our results show that SNVs alone are sufficient to identify evolutionary signatures with prognostic relevance in most cancer types, which we believe is an important data-driven result.

To avoid possible further misunderstanding and leave room for future extension of our approach, we decided to modify the name of the evolutionary signatures in "Single-nucleotide variant (SNV) Evolutionary Signatures".

Finally, it is important to note that ASCETIC does not rely on pre-annotated driver mutations as input data. Instead, ASCETIC operates on the premise that the accumulation of passenger mutations during cancer progression may occur randomly among different patients. However, a small subset of driver genes, responsible for driving tumor evolution, may exhibit consistent ordering across multiple patients. Therefore, ASCETIC identifies these small sets of genes that consistently appear in a specific order, which can be considered as driver genes. By definition, the repeated evolutions inferred by ASCETIC involve these driver genes, shedding light on their crucial role in cancer development and progression.

We acknowledge that this is a crucial point that was not adequately addressed in the previous version of the manuscript. We have now widely discussed it in the manuscript.

The authors present a validation of ASCETIC to show the accuracy of their method on simulated data, however this validation does not support an association of the detected evolutionary signatures with survival outcome. It is also not entirely clear whether the simulated data itself was created specifically for ASCETIC validation. I would suggest further validation of the evolutionary signatures identified. The font is also incredibly small in the supplementary figures.

Currently, computational methods for analyzing cancer evolution from cancer genomes are only available for the inference of cancer evolution models (#1 step of ASCETIC). Therefore, in the previous version of the manuscript, we focused our simulations on this task.

Such simulated data were not generated “ad hoc” for ASCETIC. Instead, they followed the methodology employed in other computational works on the same topic, such as [Ramazzotti, Daniele, et al. "CAPRI: efficient inference of cancer progression models from cross-sectional data." *Bioinformatics* 31.18 (2015): 3016-3026; Ramazzotti, Daniele, et al. "Learning mutational graphs of individual tumour evolution from single-cell and multi-region sequencing data." *BMC bioinformatics* 20.1 (2019): 1-13; Ramazzotti, Daniele, et al. "Lace: inference of cancer evolution models from longitudinal single-cell sequencing data." *Journal of Computational Science* 58 (2022): 101523], among others. We provide further details on the generation of simulated data in the main text's paragraph “Performance assessment via simulations” and in the Methods Section paragraph “Tests on simulated data” of the main text, which we have now expanded to offer more comprehensive information.

In addition, we included in the revised manuscript a completely new analysis for a set of for which we had a biological ground truth and that we used to perform a set of simulations. In particular, we generated data from a glioma dataset, which consisted of three distinct evolutionary signatures and subgroups. To analyze this data, we conducted 1,000 independent simulations. In each simulation, survival data were randomly sampled from the three clusters, and evolutionary steps were randomly generated according to distributions proportional to those observed within each cluster. To determine the evolutionary signatures and subgroups, we applied the ASCETIC analysis with Cox regularized regression in each simulation run. The consistency among the experiments was evaluated using the adjusted Rand index. We have thoroughly discussed these results in the main text, demonstrating that ASCETIC can reliably detect evolutionary signatures and subgroups in simulations. Additionally, we have provided a detailed report of these findings in the supplementary materials.

These results prove that our framework is able to stratify (simulated) patients in risk groups, further proving its efficacy in controlled scenarios.

Finally, we have increased the size of the supplementary figures.

Reviewers' Comments:

Reviewer #1:

Remarks to the Author:

The authors have addressed all the points I raised. I have no further comments.

Reviewer #2:

Remarks to the Author:

First of all, I'd like to thank the reviewers for addressing all of my comments. The manuscript is substantially improved. However, I still have a number of concerns:

The authors state that TRACERx, MSK LUAD and TCGA LUAD cannot be directly compared. I'm not sure I agree with this. The authors could do a restricted analysis on e.g. early stage patients, or match for disease stage.

The authors state that they identified highly concordant results in their validation work. However, I find this analysis quite confusing. The validation seems to predominantly rely on identifying the same signatures, not demonstrating that these also hold prognostic relevance. This analysis should be done.

If the focus of the manuscript is truly to demonstrate that their approach provides a new method for exploring prognosis, I think considerably more work is needed to validate this approach.

It seems that in AML, the well documented risk factor of NPM1 -> FLT3 trajectories are driving the association of evolutionary signatures with survival. Have the authors checked whether stratifying the patients purely on an analysis of co-occurrence of mutations in NPM1 and FLT3 are sufficient to stratify patients, as opposed to the evolutionary signatures. Could stratifying patients based on co-occurrence of mutations actually improve the survival result?

I would suggest that in the section "Stratification based solely on mutations", more validation is carried out to support that the evolutionary signatures improve on co-occurrence of mutations.

Reviewer #1

The authors have addressed all the points I raised. I have no further comments.

We would like to express our gratitude to the reviewer for their constructive feedback, which has been essential in significantly improving the quality of our work.

Reviewer #2

First of all, I'd like to thank the reviewers for addressing all of my comments. The manuscript is substantially improved. However, I still have a number of concerns:

We are grateful to the reviewer for the positive feedback on our work, and we sincerely appreciate their insightful comments that have helped us to improve the manuscript.

The authors state that TRACERx, MSK LUAD and TCGA LUAD cannot be directly compared. I'm not sure I agree with this. The authors could do a restricted analysis on e.g. early stage patients, or match for disease stage.

Following the reviewer's suggestion, we have enhanced the validation of lung cancer by comparing the different datasets (pages 11 and 13 of the main text). Moreover, we have included a more comprehensive and detailed characterization of the evolutionary signatures associated with different prognoses, as requested by the reviewer in their next concern.

The authors state that they identified highly concordant results in their validation work. However, I find this analysis quite confusing. The validation seems to predominantly rely on identifying the same signatures, not demonstrating that these also hold prognostic relevance. This analysis should be done.

If the focus of the manuscript is truly to demonstrate that their approach provides a new method for exploring prognosis, I think considerably more work is needed to validate this approach.

We thank the reviewer for this valuable comment.

We have carefully addressed this concern by extending the validation analyses for each of the considered cancer types. Specifically, we now present a thorough analysis and discussion of the prognostic capabilities of the discovered signatures in all cohorts (pages 9, 11, 13 and 14).

As a result, we now show in a clearer fashion that the discovered signatures hold analogous prognostic significance in all analyzed datasets.

It seems that in AML, the well documented risk factor of NPM1 -> FLT3 trajectories are driving the association of evolutionary signatures with survival. Have the authors checked whether stratifying the patients purely on an analysis of co-occurrence of mutations in NPM1 and FLT3 are sufficient to stratify patients, as opposed to the evolutionary signatures. Could stratifying patients based on co-occurrence of mutations actually improve the survival result?

I would suggest that in the section "Stratification based solely on mutations", more

validation is carried out to support that the evolutionary signatures improve on co-occurrence of mutations.

We thank the reviewer for raising this important issue.

We conducted the additional analysis suggested by the reviewer, which involved considering the co-occurrence of mutations of NPM1 and FLT3, in addition to the single-gene mutations.

Importantly, the results of this new analysis show that considering the co-occurrence of such mutations as a feature does not improve over single-gene mutations, confirming the results already presented in Supplementary Figures 17-18 and discussed in the paragraph “Stratification based solely on mutations”.

This result demonstrates that the evolutionary patterns selected by our framework can offer valuable additional information to stratify patients in risk groups.

Notice also that applying LASSO Cox regression with numerous predictors (i.e., testing all co-occurrence) poses several known challenges. It may lead to over-regularization, instability in variable selection, and computational burden. Collinear predictors can cause a loss of valuable information, and noise from irrelevant predictors may reduce predictive accuracy. Preprocessing steps like feature selection are known to be essential for improving model performance in such scenarios. See [Hastie, T., Tibshirani, R., and Friedman, J. (2009). *The elements of statistical learning*. Springer series in statistics. Springer, New York, 11th printing, 2nd edition] and [Tibshirani, R. J. (2013). *The lasso problem and uniqueness*. *Electronic Journal of Statistics*, 7, 1456-1490].

We appreciate the reviewer's insightful feedback, acknowledging the significance of this point. Accordingly, we have enhanced the discussion in the main text to better explain the implications of adopting our approach.

Reviewers' Comments:

Reviewer #2:

Remarks to the Author:

The authors have addressed my comments.

Reviewer #2

The authors have addressed my comments.

We would like to express our gratitude to the reviewer for their great feedback.